

# A Global Evaluation of Daily to Seasonal Aerosol and Water Vapor Relationships Using a Combination of AERONET and NAAPS Reanalysis Data

Juli I. Rubin[1], Jeffrey S. Reid[2], Peng Xian[2], Christopher M. Selman[1], and Thomas F. Eck[3,4]

[1]U.S. Naval Research Laboratory, Washington, DC, 20375, USA
[2]U.S. Naval Research Laboratory, Monterey, CA, 93943, USA
[3]NASA Goddard Space Flight Center, Greenbelt, MD, 20771, USA
[4]Goddard Earth Sciences Technology and Research (GESTAR) II, University of Maryland Baltimore County, Baltimore, MD 21250, USA

*Correspondence to*: Juli I. Rubin (juli.rubin@nrl.navy.mil)

**Abstract.** The co-transport of aerosol particles and water vapor has long been noted in the literature, with a myriad of implications from air mass characterization to radiative transfer. In this study, the relationship between aerosol optical depth (AOD) and precipitable water vapor (PW) is evaluated to our knowledge for the first time globally, at daily to seasonal levels using approximately 20 years of AERONET observational data and the 16-year NAAPS reanalysis v1.0 (NAAPS-RA) model fields. The combination of AERONET observations with small associated uncertainties and the reanalysis fields with full global coverage is used to provide a best estimate of the seasonal AOD and PW relationships, including an evaluation of correlations, slope, and PW probability distributions for identification of statistically significant differences in PW for high AOD events. The relationships produced from the AERONET and NAAPS-RA datasets were compared against each other and showed consistency, indicating that the NAAPS-RA provides a realistic representation of the AOD and PW relationship. The initial analysis is then extended to layer AOD and PW relationships for proxies of the planetary boundary layer, and lower, middle and upper free troposphere. It was found that the dominant AOD and PW relationship is positive, supported by both AERONET and model evaluation, which varies in strength by season and location. These relationships were found to be statistically significant and present across the globe, observed on an event by event level. Evaluations at individual AERONET sites implicate synoptic-scale transport as a contributing factor in these relationships at daily levels. Negative AOD and PW relationships were identified and predominantly associated with regional dry season timescales in which biomass burning is the predominant aerosol type. This is not an indication of dry air association with smoke aerosol for an individual event, but is more a reflection of the overall dry conditions leading to more biomass burning and higher associated AOD values. Stronger correlations between AOD and PW are found when evaluating the data by vertical layers, including boundary layer, lower/middle/upper free troposphere (corresponding to typical water vapor channels), with the largest correlations observed in the free troposphere-indicative of aerosol and water vapor transport events. By evaluating variability between PW and relative humidity in the NAAPS-RA, hygroscopic growth was found to 1) amplify positive AOD-PW relationships, particularly in the mid-latitudes; 2) diminish negative relationships in dominant biomass burning regions; and 3) leads to statistically insignificant changes in PW for high AOD events in ocean regions. The importance of hygroscopic growth in these relationships indicates that PW is a useful tracer for AOD, but not necessarily as strongly for aerosol mass. Synoptic-scale African dust events are an exception where PW is a strong tracer for aerosol shown by strong relationships even with hygroscopic affects. Given this results, PW can be exploited in coupled aerosol and meteorology data assimilation for AOD and the collocation of aerosol and water vapor should be carefully taken into account when evaluating radiative impacts of aerosol, with the season and location in mind.



**DISTRIBUTION STATEMENT A. Approved for public release. Distribution is unlimited.**
**1.0- Introduction**
The definition of an aerosol is that it is a colloidal system of particles or droplets suspended in a dispersed gaseous
medium (American Meteorological Society, 2020). While the word "aerosol" is often taken to represent only the
particulate phase, the true definition reminds us of the thermodynamic, compositional and radiative "whole" that
makes up the particulate and dispersed phases of an aerosol parcel. With this definition in mind, an important aspect
of aerosol parcels that should be considered is the covariability between the aerosol particles and the dispersed water
vapor. While the aerosol and water vapor relationship is generally accounted for in the context of relative humidity,
hygroscopicity, and optical properties (e.g., Hänel et al., 1976; Charlson et al. 1992), the covariability of aerosol
particles and dispersed water vapor is important in its own right. Early studies of co-located aerosol and water vapor
measurements demonstrated the structural covariability between the two components (e.g, Stull and Eloranta 1984;
Kleinman and Daum, 1991; Turner 2002; De Tomasi and Perrone, 2003). Now, coupled aerosol-water vapor profiles
are commonly used to infer aerosol layer structure (e.g., Livingston et al., 2003; Reid et al., 2003; 2008; 2019; Wang
et al. 2012; Yufeng et al., 2018), cloud detrainment (Su et al., 2011; Reid et al., 2019; He et al., 2021) and mixed layer
properties (Späth et al., 2016). Even integrated aerosol optical depth (AOD) and precipitable water vapor (PW)
comparisons have utility and have been used to identify air masses, transport pathways, and aerosol optical properties.
Regional studies include Africa (Kumar et al., 2017, Xian et al. 2020), the Amazon (Kaufman and Frasier, 1997;
Martins et al., 2018), India (Kumar et al., 2013; and Kannemadugu et al., 2015), and North America (O'Neil et al.,
1993; Smirnov et al., 1994). Notable examples of co-transport of aerosol particles and water vapor include the African
Monsoon Multidisciplinary Analysis (AMMA) in which elevated biomass burning aerosol layers were found with
higher water vapor concentrations than the surrounding air (Kim et al. 2009). Likewise, Marsham et al., 2016
investigated water vapor enhancements with dust in the Saharan Air Layer (SAL).
It should be noted that higher PW amounts are typically associated with higher cloud cover fractions. These higher
cloud fractions create additional environmental conditions for enhancements of the aerosol AOD and PW relationship.
There is a high RH halo around cumulus clouds (Radke and Hobbs, 1991; Perry and Hobbs, 1996) which increases
the near cloud hygroscopic growth of aerosol. Additionally, the passage of aerosol through clouds by convection
and/or advection also increases hygroscopic growth. Cloud processing of particles in cloud droplets and new particle
formation from gas-to-particle reactions in cloud water droplets are also important. Examples of remote sensing
observations from AERONET of cloud processing increasing AOD in layer clouds and/or fog are given in Eck et al.
(2012) and in the vicinity of cumulus clouds in Eck et al. (2014). Additionally high AOD events were often found to
be associated with clouds in East Asia (Eck et al., 2019; Arola et al., 2017).
In addition to its utility as a tracer for transport and mixing, the aerosol particle-water vapor co-transport are significant
in regards to their relative contributions to overall solar and terrestrial radiative effects (Rosario et al., 2011; Marsham
et al., 2016; Deaconu et al., 2019; Gutleben et al., 2019; Granados-Muñoz et al., 2019; Zhu et al., 2019; Yu et al.,
2021). Similarly, co-transport must be considered in atmospheric correction of land, ocean and atmospheric products



(e.g., Sobrino et al. 1993; Eck and Holben 1994; DeSouza-Machado et al., 2006; Luo et al, 2019; Zeng et al., 2017;
Patadia et al., 2018; Frouin et al. 2019, Ibrahim et al. 2019; Miller et al. 2019). As previously noted, there are also
links to cloud development and potentially indirect effects (Ten Hoeve et al., 2011; Pistone et al., 2016). Ultimately,
the coupled aerosol particle- water vapor system must be considered jointly to adequately contain overall climate
budgets and forcing (Kaufman and Fraser, 1997; Wong et al., 2009; Schneider et al, 2010; Sherwood et al, 2010;
Haywood et al., 2011; Huttunen et al., 2014; Yu et al., 2014; Spyrou 2018).

Finally, recent advances in coupled data assimilation (DA) allow for not only a joint analysis of aerosol particles and
water vapor as is done in weakly coupled approaches, but for observations to jointly influence posteriors through
cross-covariances in strongly coupled DA (Liu et al., 2011; Lee et al., 2017; Ménard et al., 2019). The hope is that
strongly coupled DA can be used to generate a more consistent representation of coupled atmospheric systems. In the
context of the aforementioned references on the coupled aerosol particle-water vapor system, there is now a more
pressing need for evaluating joint aerosol and water vapor measurements. This is further emphasized by the observed
frequency of aerosol and water vapor co-transport in both forecast models and satellite observations. Similar spatial
patterns between aerosol optical depth (AOD) and precipitable water vapor (PW) can be observed on a daily basis in
model analyses, forecasts, and satellite products such as the Morphed Integrated Microwave Imagery at CIMSS - Total
Precipitable Water (MIMIC-TPW) (Wimmers et al. 2011), particularly associated with mid-latitude fronts. An
example of forecasts of TPW and AOD from the Navy Global Environmental Model (NAVGEM) (Hogan et al. 2014)
and Navy Aerosol Analysis Prediction System (NAAPS; Lynch et al. 2016), respectively are shown in Figure 1 in
which co-transport regions are highlighted. Aerosol and water vapor relationships are not expected to be universal and
will likely vary in magnitude from air mass to air mass due to differences in sources, physics, and overall vertical
distribution. While the previously mentioned studies have found relationships between aerosol and water vapor for a
host of case or local studies, this relationship has not to our knowledge been evaluated on a larger spatial and temporal
scale for broad applicability for aerosol forecasting and data assimilation.

This is the first of several studies developing coupled data analysis and assimilation of the water vapor- aerosol particle
system. Here, the project begins by focusing on observations of synoptic scale temporal and spatial relationships using
the extensive NASA Aerosol Robotic Network (AERONET; Holben et al., 1998; Giles et al., 2019). The advantage
of AERONET for this study is that the data record is long and includes high frequency ground-based measurements
of both aerosol in the form of AOD and water vapor in the form of PW with sites located across the globe. Additionally,
AERONET measurements are made throughout the entire daylight hours when the sun is not obscured by clouds. It
should be noted that this does result in a high pressure bias in AERONET data since few measurements are possible
in extensive cloud fraction conditions. Another important advantage is that the observations can be made effectively
in the near vicinity of clouds without the commonly observed satellite measurement artifacts of multiple scattering
between clouds, molecules, and particles which enables a minimization of cloud contamination in the near vicinity of
clouds as compared to satellite observations. While the AERONET network is extensive, it cannot provide a full global
evaluation of the aerosol and water vapor relationship. Therefore, the relationships identified in the AERONET dataset
are compared against model AOD and PW relationships found in the NAAPS reanalysis (NAAPS-RA) dataset (Lynch



et al. 2016). A description of both the AERONET and NAAPS-RA datasets and the analyses conducting for
quantifying the global AOD and PW relationships are described in the Methods section below. The results of the
analysis are discussed in the context of large-scale relationships between column-integrated AOD and PW. A follow-
on study will then take the relationships found in this work and move on to evaluate the relationships on an event level
in space and time as well as the controlling factors that drive the aerosol and water vapor relationship, in particular,
how much synoptic scale transport controls the observed covariability.

**2 Methods**
In order to evaluate the relationship between column-integrated aerosol and water vapor in space and time, the
AERONET observational network is used as it provides joint measurements of aerosol and water vapor with low
levels of uncertainty, has a large number of sites located across the globe, and a long data record. While AERONET
measurements are column-integrated, they provide a good starting point for understanding the observed aerosol and
water vapor relationships at locations across the globe. As a first step, relationships are quantified at AERONET sites
between daily-averaged AOD and PW measurements. The focus here is on the synoptic-scale relationships between
aerosol and water vapor. Therefore, daily-averaged relationships are evaluated in this analysis, using correlations and
an evaluation of the water vapor probability distributions to identify statistically significant changes in PW with AOD.
The evaluation is then extended to the NAAPS-RA dataset in order to provide a more complete global perspective in
the full column as well as in different vertical components of the atmosphere, including the boundary layer and free
troposphere, as a means to understand how these relationships vary when considering vertical position. Finally, the
impact of relative humidity and hygroscopic growth covariability on model predicted AOD and PW relationships is
evaluated.
**2.1 Data Description:**
**2.1.1 AERONET AOD**
AERONET is a global ground-based network of sun photometers that measure direct sun and sky radiances over a
range of wavelengths (340-1640nm). These measurements are used to generate column-integrated aerosol properties
of AOD and aerosol microphysical and radiative properties (Holben et al. 1998; Giles et al., 2019). The network
includes over 600 sites with data available at https://aeronet.gsfc.nasa.gov/. The uncertainty in AERONET AOD is
reported to be ~0.01–0.02 for level 2 data with the higher uncertainty of 0.02 pertaining to the UV wavelengths and
the lower ~0.01 uncertainty associated with visible and near infrared wavelengths (Eck et al., 1999). Due to this low
uncertainty, AERONET AOD observations are used for validation of satellite retrievals (Remer et al. 2002, Ichoku et
al. 2002, Kahn et al. 2005) as well as for verification of model forecasts (Zhang et al. 2008. , Benedetti et al. 2008,
Sessions et al. 2015, Xian et al. 2019). For this analysis, AERONET version 3 (Giles et al. 2019), level 2 daily-
averaged AOD observations are used. AERONET AOD observations at 675nm for all available sites were collected
and sites that had a minimum of 100 daily-averaged values were retained for the analysis, with seasonal data counts
in Figure 2. The 675nm wavelength was selected as it is a core AERONET wavelength that is available at all sites and
provides parity for both the fine and coarse aerosol modes.






### 2.1.2 AERONET Water Vapor

Precipitable water vapor (PW), a measure of the total amount of water vapor contained in a vertical column from the
surface to the top of the atmosphere, is retrieved from AERONET direct sun irradiance measurements in the water
vapor absorption band around 940nm. The uncertainty of AERONET water vapor data is reported at 12% (Sano et al.
2003) and more recently, an analysis of uncertainty against radiosonde, microwave radiometry, and GPS data indicated
a dry bias of 5-6 % and a total estimated uncertainty of 12-15% (Perez-Ramirez et al. (2014)). The Perez-Ramirez et
al. (2014) evaluation with the identified uncertainty range of 12-15% included PW retrieval comparison at 3 sites
located in the tropics, mid-latitudes and the arctic, covering a range of climatic conditions and temperature/water vapor
profiles, and therefore, provides a reasonable uncertainty estimate for the entire AERONET network. The PW data
used in this analysis comes from the same AERONET version 3, level 2 daily-averaged dataset that is used for the
AOD data. As was the case for the AERONET AOD data, sites that had a minimum of 100 daily-averaged values
were retained for the analysis (Figure 2).

### 2.1.3 NAAPS Reanalysis

The NAAPS aerosol reanalysis v1.0 (Lynch et al. 2016) is a standardized global modal AOD product generated by
the U.S. Naval Research Laboratory (NRL) that extends over a 16 year time period (2003-2019). The core of the
aerosol reanalysis is the NAAPS offline aerosol transport model and its associated 2-dimensional variational data
assimilation system, the Navy Variational Data Assimilation System for Aerosol Optical Depth (NAVDAS-AOD).
NAAPS has been run semi-operationally at NRL since 1998 and became operational at the Fleet Numerical
Meteorology and Oceanography Center (FNMOC) in 2006 with NAVDAS-AOD operationally implemented in 2010.
For the NAAPS-RA, NAVDAS-AOD is used to assimilate quality-assured and quality-controlled AOD retrievals
from the Moderate Resolution Imaging Spectroradiometer (MODIS) and Multi-angle Imaging SpectroRadiometer
(MISR).

NAAPS generates 3-dimensional forecasts of dust, smoke, sea salt and a combined anthropogenic/biogenic fine
aerosol (ABF, also referred to in this work as pollution) mass concentration fields and the associated 3-dimensional
aerosol extinction and column-integrated AOD fields. As on offline model, NAAPS is driven by meteorological fields
from the Navy Global Environmental Model (NAVGEM) (Hogan et al. 2014), using analysis fields every 6 hours and
forecasts provided at 3 hour intervals. The NAVGEM analysis fields are generated using NAVDAS for assimilation
of a large number of conventional and satellite-based observations (Daley and Barker, 2001). NAVGEM variables
used by NAAPS includes the topography, sea ice, snow cover, surface stress, surface heat/moisture fluxes,
precipitation, lifting condensation level, cloud cover and height as well as 3-dimensional winds, temperature, and the
most relevant for this work, humidity. For the NAAPS analysis, aerosol sources, including dust and smoke, and
deposition processes were regionally tuned to best match observations (AERONET, MODIS). A detailed description
of the NAAPS-RA v1.0 is described in Lynch et al. 2016.



In NAAPS, the Hanel (1976) formulation of the hygroscopic growth factor (f) for a given species i and relative
humidity (r) is used to represent the effect of humidity on particle light scattering, defined as:
$$f_i(r) = \left[\frac{(1-r)}{(1-r_o)}\right]^{-\gamma_i},$$  (1)
where $\gamma_i$ is an empirical species-dependent exponent (ABF=0.5 assuming 40% sulfate and 60% organics, smoke=0.18,
sea salt = 0.46, dust=0) and $r_o$ is the reference relative humidity of 30%. The hygroscopic growth factor is applied
when calculating the aerosol scattering coefficient. In order to assess the impact of hygroscopic growth on model-
predicted AOD and PW correlations, a "dry" AOD is also calculated for the NAAPS-RA in which the hygroscopic
growth factor is not applied.
For this work, the NAAPS-RA v1.0 AOD fields and the NAVGEM humidity fields used in generating the NAAPS-
RA are extracted for the full 16 year dataset (2003-2019). NAVGEM humidity fields were integrated vertically to
generate model-predicted PW fields and both the PW and AOD fields were averaged on a daily basis. Additionally,
"dry" AOD fields were calculated for the NAAPS-RA and likewise, averaged on a daily basis.
**2.2 AOD and PW Relationship Analysis:**
**2.2.1 Correlation Analysis**
As a first step in understanding the relationship between aerosol and water vapor in the AERONET data record,
correlations (Pearson correlation coefficients) are calculated at each AERONET site with a minimum of 100 data
points. The correlations are calculated between the daily-averaged AOD (675nm) and PW datasets seasonally
(December-January-February (DJF), March-April-May (MAM), June-July-August (JJA), September-October-
November (SON)). This analysis is used to identify when and where relationships exist between AOD and PW in the
data record and the strength of the relationship. In order to provide global context to the AERONET AOD and PW
correlations, the same analysis was conducted using the NAAPS 16-year v1.0 reanalysis dataset. The seasonal
reanalysis correlations were calculated in a similar manner as the AERONET data, using daily-averaged model-
generated AOD and PW values. The model-generated values were then compared against observationally generated
AERONET correlations. The correlations in both the AERONET and NAAPS-RA evaluation were tested for
statistical significance at the 95% confidence level.
**2.2.2. Slope Evaluation**
In addition to the AOD and PW Pearson correlation coefficients calculated from the AERONET and NAAPS-RA
datasets, the slopes of the AOD and PW relationship were calculated from the seasonal data using a Theil-Sen
regression. A Theil-Sen regression is a robust method for fitting a line to sample points by choosing the median of
slopes of all lines through pairs of points. Due to the use of the median slope, the Theil-Sen method is insensitive to
outliers and therefore, a useful method for this analysis. With the Theil-Sen regression, a 95 % confidence interval of
the Theil-Sen slope was calculated for each location and season.
**2.2.3 Evaluation of the AOD and PW Probability Distribution**





In addition to a correlation and slope analysis, the AOD and PW probability distributions were also evaluated. Given
the expectation that aerosol and water vapor relationships will change depending on the air mass, seasonal correlations
can obscure the presence of aerosol and water vapor relationships when air masses with an existing relationship
between aerosol and water vapor occur infrequently. In this evaluation, the PW distribution associated with high AOD
events, defined as having an AOD value greater than 1 standard deviation above the mean, were compared to the PW
distribution for all data for a given location and season. A t-test was conducted to identify statistically significant
differences in the PW distribution means for the high AOD events and all data (p-value = 0.05). This analysis was
conducted seasonally (DJF, MAM, JJA, SON) using both the AERONET and the NAAPS-RA datasets.

### 2.2.4 Vertical Evaluation of the AOD and PW Relationship

While the global and AERONET site AOD and PW evaluations as well as the studies cited in the introduction provide
an understanding of the column-integrated relationship between aerosol and water vapor, an additional evaluation was
conducted to look at the aerosol and water vapor relationship in different levels of the troposphere. This evaluation
was conducted using the NAAPS-RA fields only, since observations of joint aerosol and water vapor vertical structure
are limited. Model generated correlations were calculated for a defined boundary layer (BL), lower free troposphere
(LT), mid-free troposphere (MT), and upper free troposphere (UT) region. The reanalysis total aerosol extinction and
specific humidity were vertically integrated in the first 1km of the atmosphere as a representation of the boundary
layer. Integration levels in the free troposphere were selected based on the sensitivities of the upper, mid-level, and
lower-level geostationary water vapor channels on the NOAA Geostationary Operational Environmental Satellite
(GOES) Advanced Baseline Imager (ABI) and the JMA Advanced Himawari Imager (AHI) with a goal of using these
water vapor channels to further explore aerosol and water vapor relationships in future work. The selected integration
levels were from 800 to 500 hPa (LT), 600 to 300 hPa (MT), and 400 to 300 hPa (UT), respectively. The vertically
integrated relationships, as was done for the full column-integrated evaluation, are calculated seasonally and are used
to identify if the model correlations are controlled by aerosol and water vapor in certain parts of the atmosphere. While
the boundary layer is expected to be a dominant control of the signal, given the sources of both aerosol and water
vapor are within the boundary layer, strong correlations within the free troposphere could indicate aerosol and water
vapor relationships as a result of lifting from the surface or long-range transport which typically occurs within the free
troposphere.

### 2.2.5 Impact of Hygroscopic Growth on AOD and PW relationships

It is well documented in the literature that water uptake on aerosol particles under moist conditions impacts aerosol
optical properties. Because of this, it is necessary to understand how much hygroscopic growth impacts AOD and PW
relationships through covariability of PW and relative humidity. The data to evaluate this observationally is not
available, therefore, the NAAPS-RA is used to evaluate the impact of the hygroscopic growth factor on model
predicted correlations. As a first step in this evaluation, the correlation between PW and relative humidity was
calculated by season for the previously defined vertical components of the atmosphere (boundary layer,
lower/mid/upper free troposphere). In order to calculate relative humidity for each defined part of the troposphere, a
saturation specific humidity was calculated in each model level using the reanalysis pressure and temperature fields





as input. Both the specific humidity and the saturation specific humidity were vertically integrated over the defined
levels and the ratio of the two values was used to produce a relative humidity that conserves the amount of water vapor
through the associated portion of the troposphere. This analysis gives a first look at where the covariability between
PW and RH is expected to be most impactful on the AOD and PW relationship. However, given aerosol hygroscopic
growth is dependent on aerosol type, the analysis was taken a step further by calculating the seasonal relationships,
including correlations/slopes and the probability distribution evaluation, between dry AOD and PW. The dry AOD,
in which the impact of hygroscopic growth on AOD is removed as described in the NAAPS-RA section (2.1.3), was
calculated for the full dataset. The relationships using the dry AOD are compared to the standard AOD/PW results as
a means to evaluate the impact of hygroscopic growth on the modeled AOD and PW relationships.

### 2.2.6 Evaluation at Individual AERONET Sites

While the previous analyses provide a global perspective on the aerosol and water vapor relationships, the relationships
were also evaluated at select AERONET sites to provide a first look at what is driving the observed covariability
between AOD and PW on an event level. The AERONET sites, including Tallahassee, Florida in the Southeast United
States, Beijing, China in East Asia, Izana, Canary Islands off the coast of Africa, and Alta Floresta, Brazil in South
America, are selected based on the strength of the observed/modeled relationships and cases are selected for different
seasons that exhibited both positive and negative relationships. While this evaluation does not by any means provide
a complete understanding of the drivers of these relationships across the globe, it can be used to provide some insight.

### 3.0 Results

This study is highly multi-dimensional. In order to elucidate the findings, the results are first presented as a global
evaluation, which is followed by a more in-depth discussion by region, level, and accounting for the impacts of
hygroscopicity. As aerosol regimes are typically seasonal in nature, all evaluations are performed for DJF-MAM-JJA-
SON. Summaries of the data used in the analyses are presented in Figures 2 through 6, including: Figure 2, seasonal
counts of daily-averaged AERONET AOD and PW data; Figure 3, seasonal mean AERONET and NAAPS-RA AOD
and PW values; Figure 4 seasonal NAAPS-RA AOD averages by aerosol types (dust, sea salt, anthropogenic/biogenic
fine, biomass burning); and Figures 5 and 6, the NAAPS-RA AOD and PW (respectively) 25th, 75th, and 90th
percentiles from daily data and associated interquartile range (IQR) of by season. In regards to the AERONET
analysis, only sites with a minimum of 100 data points are included, as previously discussed. Due to this constraint,
some temporary sites used for field campaigns are excluded in this work.

### 3.1 Global Patterns of AOD-PW Correlation

Overall, the seasonal patterns in both AOD and PW are pretty consistent between AERONET and the NAAPS-RA
(Figure 3). For example, peak AOD values in North America and Europe occur during the summer months in both
datasets. Likewise, peak AOD values are found over the Sahel in winter and spring due to a combination of dry-season
biomass burning and dust associated with the northeasterly Harmattan winds with shifts in peak AOD further north in
summer due to increased dust activity over the Sahara. Like the Sahel, peak AOD values associated with fire activity
during regional dry seasons are also found in both datasets for Central and South America, Southern Africa and



Southeast Asia. Boreal regions, which also exhibit seasonality due to fires in summer months, are not as well sampled in the AERONET dataset, making it harder to see seasonal shifts in AOD. However, this seasonality is found in the NAAPS-RA. Likewise, northward shifts in PW are seen in AERONET and the NAAPS-RA in the summer and a southward shift in the winter months. A more in-depth discussion of the data by region is below:

1) North America: The largest number of AERONET sites are present in this region with ~180 included in the analysis. AERONET data counts are the highest in the summer months (JJA) which also coincides with peak mean AOD and PW values in both the AERONET and NAAPS-RA datasets (Figure 3). Summertime peak AOD values are associated with ABF and smoke aerosol types, concentrated to the North, and a combination of ABF and transported dust to the South (Figure 4). Despite JJA being associated with the highest AOD values, the IQR is only around 0.1-0.2 (Figure 5). The 90th percentile AOD values in JJA for North America are mainly associated with large smoke events, particularly originating from the Pacific Northwest and Boreal regions (Figure 5). High AOD values are also observed in MAM months concentrated in the Southeast United States (Figures 2 and 5), associated with smoke (originating from Central American fires) and ABF/pollution aerosol types (Figure 5).

2) Europe: Data from ~125 AERONET sites was included in the analysis in Europe. Like the North America region, peak AERONET data counts occur during JJA months (Figure 2). Peak AOD values are observed during MAM and JJA (Figures 3 and 5), mainly associated with pollution in Eastern Europe, and Mediterranean dust (Figure 4). PW values also peak during JJA (Figures 2 and 6). AOD IQR values, like North America, are relatively small and on the order of 0.1-0.2 in JJA/MAM, with 90th percentile AOD events in the 0.3-0.5 range.

3) East Asia: The analysis in East Asia included data from ~52 AERONET sites. AERONET data counts are relatively consistent throughout the seasons (Figure 2). AOD values in East Asia are high throughout the year due to the presence of pollution, concentrated to the East and dust, particularly in the spring and summer (Figure 4). While pollution aerosol is present throughout the year, AOD values tend to be higher in the winter months than the summer months in the NAAPS-RA (Figures 3 and 5) with the strength of the East Asian Monsoon being a controlling factor in the spatial distribution and aerosol concentration in the region (Zhang et al. 2010; Yan et al. 2011; Zhu et al. 2012; Mao and Liao, 2017). However, in the AERONET dataset, the highest AOD values are observed in the summer months, consistent with the literature (Eck et al. 2005, 2018). This discrepancy may be related to the satellite data that is assimilated in the NAAPS-RA in the summer months. High AOD values are often misclassified as cloudy by the retrieval algorithms and subsequently screened (Eck et al. 2018), which can contribute to low AOD biases in the model (Reid et al. 2022). The range in AOD values is particularly large over East Asia as shown by the percentiles in Figure 5 with peak IQR values of around 0.6-0.7 occurring during DJF.

4) South America: Data from ~44 AERONET sites was used in the analysis in South America. AERONET data counts are the greatest during JJA and SON months, which is coincident with the highest AOD values. This is particularly the case in SON, which is the dry season in South America when fire activity is increased.





331  The dominance of smoke aerosol is shown in the NAAPS-RA for these months (Figure 4). Extreme event
332  AOD values (90th percentile) and the IQR are the greatest for SON, again due to fire activity (Figure 5).

333 5) Northern Africa: Data from ~39 sites were used for evaluation in Northern Africa.  Data counts are relatively
334  consistent across the seasons with the exception of the Banizoumbou, Niger site with approximately 1600
335  data points from 16 years of data during the DJF season. The AERONET and reanalysis average AOD values
336  for the North African Sahel region peaks in the winter and spring months (DJF, MAM) due to a combination
337  of dust and smoke aerosol (Figure 4). Peak Sahel AOD values coincide with the ITCZ being its most southern
338  position, which is shown in the PW fields (Figure 3 and 6). North Africa, particularly the Sahara, has high
339  AOD in the spring and summer months due to dust outbreaks with peak AOD values exceeding 1 and IQR
340  values in the 0.4-0.5 range (Figure 5).

341 6) Southern Africa: The analysis in Southern Africa included data from ~30 AERONET sites.  AERONET data
342  counts are pretty consistent throughout the year, however, there are less sites available for analysis during
343  the DJF months. AOD values in Southern Africa are the highest in JJA and SON which is coincident with
344  peak fire activity in the region.

345 7) Arabian Peninsula: AERONET data counts from ~20 sites are consistent across the seasons in this region.
346  While dust emissions are present through the year, peak dust activity occurs in the summer months as shown
347  in the AERONET and NAAPS-RA AOD mean and percentile values (Figures 3 and 5).

348 8) India: The number of AERONET sites was ~20 in India with locations concentrated towards the North for
349  sampling the Indo Gangetic Plain in which pollution dominated AOD is present throughout the year, with
350  peak AOD values exceeding 1 during all seasons (Figure 3-5). Dust aerosol from the Thar Desert and the
351  Arabian Peninsula are transported to western India, particularly in the MAM and JJA seasons, while smoke
352  aerosol contributes to AOD in eastern India in MAM. AOD and PW are heavily influenced by the summer
353  monsoon season in which peak PW is observed (Figures 3 and 6).

354 9) Southeast Asia: Data from ~21 sites was available for the analysis in Southeast Asia.  The number sites used
355  is greatest in the spring (Figure 2), coincident with the Peninsular Southeast Asia fire season in which peak
356  AOD values exceed 1 and large IQR values are present (Figure 5). Peak AOD values shift towards Insular
357  Southeast Asia during the SON months in which fire activity increases. Pollution is also present throughout
358  the year.

359 Regressions of AOD and PW for the daily data by season, including correlation coefficients and slopes, and the
360 statistically significant difference in mean PW between the distribution associated with high AOD events only and the
361 full PW distribution for both the NAAPS-RA and AERONET daily data are presented in Figure 7 with confidence
362 intervals on the Theil-Sen slopes shown in Figure 8. Red regions/sites indicate a positive correlation in which higher
363 PW is associated with higher AOD values and blue regions indicate a negative relationship in which lower PW is
364 associated with high AOD values. For all evaluations in Figure 7, the predominant signal is positive (ie. red) in both
365 the AERONET observations and the NAAPS-RA with the strongest correlations varying by season, and/or aerosol
366 regime. In the AERONET dataset, the strongest positive correlations are summarized in Tables 1-4 for DJF, MAM,
367 JJA, and SON, respectively. Also included are NAAPS-RA values for these sites as a means of comparison. For winter



months (DJF), the strongest positive correlations (>0.6) occur at sites in the Southeast United States, East Asia and
select sites in the Middle East such as Dhadnah, UAE (Table 1). In the spring months (MAM), dominant positive
relationships occur at mostly Eastern United States sites and the Nainital site in India (Table 2). Southern Africa sites
associated with smoke aerosol, Eastern European sites, and select sites in the Eastern United States have the strongest
positive correlations in the summer months (JJA) (Table 3, Figure 7). In the fall (SON), AERONET positive
correlations are strongest for the Eastern United States, select European sites as well as a site at Dhadnah, United Arab
Emirates.
The NAAPS-RA daily correlations (Figure 7) within seasonal aggregates indicate similar but not identical spatial
patterns relative to the AERONET dataset. The dominant positive correlation regions include the Eastern/Southeastern
United States as is found in AERONET. Likewise, stronger European AOD and PW correlations are found in the
summer months, in Eastern Asia in the winter season, and the Middle East in the fall. The NAAPS-RA results are
helpful in that it provides a more complete perspective on the AOD and PW relationships. In addition to strong positive
correlations in Southeast United States and East Asia during DJF, the NAAPS-RA also indicates strong positive
correlations in parts of Southwest Asia (Iran/Afghanistan/Pakistan), India, South America, and Southern Africa, which
are minimally if not at all sampled by AERONET. The spatial extent of the observationally-sampled relationships can
also be seen. For example in MAM, the AERONET correlation at the Tamanrasset site in Algeria is 0.55 with a
consistent NAAPS-RA correlation of 0.53. In the reanalysis, the correlations, greater than 0.5, extend to the east of
Tamanrasset. Likewise, the spatial extent of correlations for maritime regions can be seen in the reanalysis, where
AERONET sites are rare. In JJA months, correlations at the Dahkla site in Morocco are 0.45 in the AERONET dataset.
Although the NAAPS-RA correlation at Dahkla is weaker (R=0.31), the positive relationship observed in both datasets
on the West coast of Africa can be seen extending out into the Atlantic ocean in the reanalysis, consistent with dust
transport pathways. Correlations associated with aerosol transport are also seen in Southern Africa in the reanalysis,
extending out into the ocean.
Although positive correlations are dominant throughout the world, negative correlations were also identified in both
the AERONET and NAAPS-RA datasets from daily data. In the AERONET dataset, negative correlations are limited
to the tropic/subtropics with negatively correlated regions mostly associated with biomass burning. The strongest
negative correlations in the AERONET dataset are shown in Tables 1-4 with NAAPS-RA values shown for
comparison. During all seasons, negative correlations are found in the Sahel region in both AERONET and the
NAAPS-RA with the negative relationships extending further northwards in the spring and summer months. This
results in an exceptionally strong dipole between Saharan and Sahelian outflow and is likely related to shifts in the
ITCZ. This points to aerosol sources (biomass burning and dust) and scavenging as a cause of the negative AOD and
PW relationship. The NAAPS-RA shows these negative correlations extending into the Atlantic Ocean with seasonally
dependent differences. Negative correlations extend into the Caribbean in JJA and to northern parts of South America
in MAM, consistent with seasonal transport pathways. Other negative correlation regions include Southeast Asia,
South America, and Southern Africa. For these regions, the strongest negative correlations are associated with the
respective dry, burning seasons. For example, negative correlations are strongest in Peninsular Southeast Asia in





MAM and in Insular Southeast Asia and South America in SON. In these cases, negative AOD and PW relationships are likely a result of higher aerosol emission occurring under dry conditions, which lead to more fire activity. Southern Africa is an exception during JJA, in which smoke aerosol is dominant (Figure 4). However, this is consistent with previous studies which have found elevated free tropospheric water vapor levels associated with Southern African smoke events (Adebiyi et al. 2015; Pistone et al. 2021). Correlations in both AERONET and the NAAPS-RA are positive in JJA and negative in MAM and SON when smoke aerosol is also present, but not at its peak. One of the largest AERONET negative correlations occurs at the Jomsom, Nepal site in JJA with a value of -0.65 (Table 3), although nearby sites show small or statistically insignificant correlations. The Jomson site is located at 2825 meters with maximum PW values around 2 while the nearby Pokhara site is 2000 meters lower in altitude with maximum PW values around 5, therefore, Jomson is likely a regional outlier due to altitude effects. For Jomsom and the surrounding regions, the NAAPS-RA indicates no statistically significant correlation. While NAAPS and AERONET are in general agreement in the locations of negative correlations, this discrepancy is likely related to meso or small scale features that are not captured in a global, 1 degree model.

### 3.2 Consistency between AERONET and NAAPS-RA

While the global plots of AERONET and NAAPS-RA AOD and PW relationships give a sense of spatial agreement, a scatterplot comparison of the quantitative values generated from the two datasets are used to take a closer look at the consistency between the observed and predicted relationships. A seasonal comparison of AERONET and NAAPS-RA regressions is shown as a scatterplot in Figure 9, including site by site a) correlations; b) Theil-Sen slopes; and c) the PW mean difference for high AOD events. In addition to the three scatterplot comparisons, all locations for which for the sign of the AOD and PW relationships differed between the AERONET and the NAAPS-RA datasets were identified. For these identified sites, the distribution of AERONET correlations are plotted by season in Figure 9d. This is included as a means to examine the strength of the observed AOD/PW relationship under conditions when the datasets disagree. Overall, the observations and model are in general agreement in the sign of the correlations (Figure 9a) with similar results found for the Theil-Sen slope and the PW mean differences for high AOD events (Figure 9b,c). Differences in the sign of the correlation are found for 15.5, 9.5, 10.2, 10.1% of analyzed AERONET sites for the DJF, MAM, JJA, and SON months, respectively. For all seasons except JJA, these differences are mostly associated with a negative correlation in the AERONET data and a positive value in the NAAPS-RA. Differences in correlation sign occur for sites in which the AERONET-generated correlations are weak, mostly falling below 0.20 (Figure 9d), with the exception of the Jomson AERONET site in JJA in which AERONET indicated a strong negative correlation and the reanalysis had a slight positive, but statistically insignificant relationship as previously discussed. For the strongest correlation sites, AERONET and NAAPS are in good agreement in DJF and MAM (Tables 1 and 2). For JJA and SON, NAAPS-RA has a tendency to produce weaker correlations relative to AERONET (Tables 3 and 4). Some differences are expected given that the event sampling is different between the AERONET observations and the 16-year NAAPS-RA. However, the overall agreement in the correlations between the two datasets provides some confidence in the NAAPS-RA for generating regionally and seasonally varying AOD and PW relationships on a global scale.





### 3.3 Slope Evaluation

With the consistency between AERONET and NAAPS-RA established, a more thorough evaluation of the strength of the slope of AOD-PW relationship has been conducted. As previously discussed, the AOD and PW relationship in both the AERONET and NAAPS-RA datasets were quantified using a Theil-Sen regression in order to fit a slope to the change in AOD per unit cm PW (Figures 7 and 8). Examples of NAAPS-RA and AERONET Theil-Sen fittings for eight AERONET sites scattered over the globe, each with their own unique aerosol environment are shown in Figure 10. Included are positive and negative correlation examples shown for each season (DJF: Beijing China and Lamto, Ivory Coast; MAM-Houston Texas and Ilorin Nigeria; JJA-Helsinki, Finland and Dakar Senagal; and SON-Dhadnah UAE and Palangkaraya, Indonesia) with the selected sites having some of the strongest correlations for the respective seasons in the AERONET and NAAPS-RA datasets (Tables 1-4). Good agreement is shown between the NAAPS-RA and AERONET-generated Theil-Sen slopes at the selected sites with the largest differences occurring at the Lamto and Palangkaraya sites in which relatively less AERONET observations are available. These fittings are calculated for each grid and AERONET site and are used to generate the results in Figures 7 and 8. The examples in Figure 10 show the insensitivity of the Theil-Sen regression to outliers, while the correlation coefficient is quite sensitive to such values. Beijing in DJF exhibits a large change in AOD with PW, as high AOD events are more frequent at this location (Figures 5 and 10). However, places like Houston, Helsinki and Dhadnah have relatively smaller Theil-Sen slopes as high AOD events, with a value around 1, occur less frequently and do not influence the slope. For these locations, the range of frequently observed AOD events is much smaller (Figures 5 and 10), resulting in small changes in AOD with PW. Although there is certainly scatter in the data points in Figure 10, statistically significant trends exist. The scatter in the data points occurs more so at negative correlation locations (Figure 10), resulting in smaller correlation coefficients. While the relationships for both positive and negatively correlated locations are statistically significant and the Theil-Sen regression gives an overall trend, the scatter indicates differences in AOD and PW relationships will occur from day to day. This is expected as the AOD-PW relationship is based on a combination of transport covariance and local meteorology-source relationships.

The global and seasonal pattern in the positive and negative Theil-Sen slopes are consistent with the correlation analysis results (Figures 7 and 8). The biggest Theil-Sen slopes tend to occur where larger IQR ranges are present (Figure 5), as was shown for the Beijing Theil-Sen slope example in Figure 10. The largest slopes in both datasets are centered on Beijing in the DJF months with values exceeding $1 cm^{-1}$. Beijing consistently has some of the largest positive changes in AOD with PW in the AERONET dataset for all seasons with values, including 95% confidence intervals, of 1.1(1.0-1.2), 0.35(0.32-0.38), 0.46(0.43-0.51), and 0.26(0.22-0.3) cm-1 for DJF, MAM, JJA, and SON, respectively. The NAAPS-RA is largely consistent with AERONET for the DJF and MAM months with corresponding values of 1.13(1.08-1.18), and 0.31(0.29-0.33) cm-1. Less sensitivity to PW is found in the reanalysis for JJA and SON with corresponding values at Beijing of 0.13(.11-0.14), and 0.18 (0.17-0.20) cm-1. This is likely due to an underestimation of haze formation within NAAPS, as with other global models (e.g., Sessions et al., 2015; Xian et al., 2019) and also possibly due to the underestimation of AOD in summer from NAAPS due to a low AOD bias in the assimilated satellite AOD datasets in the East Asia region (Eck et al., 2018). Large positive changes in AOD with PW extend through Asia, the Middle East, and Northern Africa, all regions impacted by high AOD events. As is the case





in the correlation results, the strong dipole in slopes is clear over North Africa with positive slopes to the North and
negative slopes in the southern Sahel region. Likewise, negative slopes are mainly associated with burning regions
with the exception of Southern Africa in JJA. Statistically significant correlations and slopes at high latitudes,
particularly Antarctica, indicate aerosol/water vapor transport in the model since local sources are limited, although
AOD and PW values are low (Figures 5 and 6).
Recall the scatterplot comparison of AERONET and NAAPS generated changes in AOD with PW is shown in Figure
9(b). Again, there is generally good agreement between the datasets, consistent with the correlation comparison. The
signs of the slopes are the same with the exception of 14.6, 9.2, 12.3, and 8.1% of sites for DJF, MAM, JJA, and SON,
respectively. Sites where differences in sign are observed have weak correlations (Figure 8d). The NAAPS-RA has a
tendency to under-predict negative changes in AOD with PW relative to AERONET for SON months where peak
negative slopes are generated from AERONET. This is also shown in Table 4 as well as the global maps in Figure 7
where differences can be seen, particularly in the Sahel and Southeast Asia. At the Kuching site in Borneo in SON,
the AERONET generated slope is -0.37(-0.48 to -0.28) cm-1 with a reanalysis value of -0.11(-0.13 to -0.09) cm-1
likely due to strong mesoscale variability and poor constraint in biomass burning on Borneo (Reid et al., 2013; Wang
et al., 2013). Additionally, this could again be due to satellite retrieval screening of smoke as cloud and NAAPS failing
to simulate the highest AOD smoke events in Borneo, especially in the dry El Nino years such as 2015 (Eck et al.
2019, Shi et al. 2019). The reanalysis also tends to under predict positive slopes for JJA at the AERONET sites where
the largest slopes are observed. This difference is not restricted to a particular region, but can be seen in East Asia,
Africa, and Mexico City (Figure 7). As an example, the Tamanrasset site in Algeria exhibits slopes in the AERONET
data of 0.26(0.23-0.30) cm-1 and in the reanalysis of 0.07(0.06-0.08). Likewise, at the Lubango site in Angola, the
slope is 0.27(0.21-0.34) cm-1 in the AERONET data and 0.13(0.12-0.14) cm-1 in the reanalysis. The Tamanrasset
site is at 1377 meters altitude in the Ahaggar Mountains which is significantly higher than the surrounding terrain in
the Saharan Desert. The Lubango site in Angola is at 2047 meters, also higher than a portion of the surrounding terrain.
This terrain/altitude influence is likely a factor in the discrepancies. The differences in slopes for JJA are also shown
in Table 3.

### 3.4 Evaluation of the AOD and PW Probability Distribution

While the correlation and slope evaluation is used to define a seasonal AOD and PW relationship across the datasets,
it is expected that variations in the aerosol and water vapor relationship will exist across air masses. As a result, a
probability distribution evaluation is another useful way to examine the data. The seasonal evaluation of the probability
distributions is included in Figure 7, below the correlation and slope results. The plots show the statistically significant
difference in the mean for the PW distribution associated with high AOD events (AOD values more than 1 standard
deviation above the mean) and the full PW distribution. Red regions/sites indicate that the PW mean for high AOD
events is statistically higher than the full distribution mean (i.e. higher moisture levels). Blue regions/sites indicate a
lower PW mean for high AOD events (i.e. dryer conditions). Regions or sites in white have no statistically significant
difference. The spatial pattern in the probability distribution evaluation are similar to the correlation and slope analysis,
however, the probability distribution evaluation highlights different regions than the previous analyses. For example,





across all seasons, larger changes in PW for high AOD events are observed in Argentina, South America, including
at the CEILAP-BA (Buenos Aires, Argentina) with values of 0.88, 0.94, 1.01, and 1.00 cm for DJF, MAM, JJA, SON
in the AERONET dataset and values of 0.61, 0.35, 0.95, and 0.73 cm in the NAAPS-RA dataset. This is a region that
is impacted by both local pollution and transported biomass burning (Resquin et al. 2018). Larger changes in PW for
high AOD events are also observed over Northern Australia during MAM, which is consistent with peak bushfire
season in the region. Larger changes in PW are also found over the United States and Canada, consistent with patterns
in the correlation evaluation, but with more pronounced values relative to other locations. ABF is generally the
dominant aerosol type with biomass burning from Central America and Western US/Boreal Regions during the MAM
and JJA seasons, respectively. Likewise, Eurasian Boreal regions associated with biomass burning activity during JJA
are more pronounced in the PW distribution evaluation. The peak in values in the Southeast United States are found
during the DJF season. During MAM and SON, the peak areas include most of the Eastern United States and extending
into Canada and Central America. However, regions that were more pronounced in the correlation and slope
evaluation, have smaller differences in mean PW for high AOD events, Beijing being a good example of this. Based
on AERONET, the difference in mean PW at Beijing is 0.21cm while the difference is 0.35cm in the NAAPS-RA for
DJF when the strongest correlations and largest slopes were found. However, sites like Stennis, Mississippi (Table 1)
which had a much smaller slope than Beijing (0.04cm-1 compared to 1.1cm-1) have a much larger difference in mean
PW with the AERONET value of 1.08cm and the NAAPS value of 1.16cm. This is because the probability distribution
evaluation is taking into account those infrequent, outlier events that don't affect the Theil-Sen slopes. Locations
where the IQR is relatively small, such as the United States, Europe, Australia, and parts of South America and
Southern Africa have greater differences in mean PW, despite having small Theil-Sen slopes, due to the impact of
outlier events. For many of these regions, the outliers are associated with biomass burning, indicating that PW is a
useful tracer for such events.
Like the correlation and slope evaluation, a comparison of AERONET and NAAPS generated differences in mean
PW was conducted by season. Similar to the previous two comparisons, AERONET and NAAPS are in agreement in
the sign of PW difference for most locations, demonstrated by the global plots in Figure 7 and the scatterplots in
Figure 8c. The % of sites that have differences in sign between the two datasets are 8.9, 6.87, 5.86, and 4.15% for
DJF, MAM, JJA, and SON, respectively. These percentages are smaller than the % of sites with differences in the
correlation and slope analysis. However, like the previous evaluations, most sites that exhibit sign differences between
AERONET and NAAPS had weak AOD and PW relationships (R<0.20, Figure 8d), with the exception of some
outliers in which small scale features that cannot be resolved in the global model may be at play. The comparisons
between AERONET and NAAPS-RA across the different evaluations indicate that NAAPS is generating AOD and
PW relationships that are pretty consistent with the observational data. Although differences in magnitude are present,
the direction of the relationships are very consistent, providing confidence in the use of the NAAPS-RA for further
exploring the AOD and PW relationship, particularly in the vertical and accounting for hygroscopic affects.
**3.5 Vertical Evaluation of the AOD and PW Relationship**


In addition to calculating the full column-integrated AOD and PW correlations in the NAAPS-RA, the correlations
were also evaluated by vertically integrating the extinction and specific humidity through previously defined pressure
levels in the atmosphere that correspond to a boundary layer, lower, middle, and upper free troposphere. This
evaluation was conducted seasonally, like the fully integrated analysis, with results shown in Figure 11. In addition to
the global plots, histograms of the AOD and PW correlations for the full column and the vertical components of the
atmosphere are shown in Figure 12. It is notable that stronger positive correlations exist when looking at limited parts
of the atmosphere compared to the fully integrated column. This is most evident in the global plots for ocean regions,
particularly in the Southern Hemisphere, where correlations exceeding 0.5 occur compared to the fully integrated
correlations that are on the order of 0.2. This result is not unexpected given that the vertical components of the
atmosphere look different depending on things like vertical mixing, a local aerosol and water vapor source compared
to a long-range transport event, the relative humidity profile etc. Additionally, some regions exhibit stronger
correlations in certain portions of the atmosphere. For example, dust dominated regions such as the Sahara, Arabian
Peninsula and the Gobi and Taklimakan deserts have the strongest correlations in the mid FT. This is higher up in the
atmosphere than expected, given for example, studies have shown East Asian dust heights to range from 1.9 to 3.1km
(Liu et al. 2019) and the typical description of the Saharan Air Layer (SAL) includes dust-laden air between
approximately 850 and 500 hPa (Karyampudi et al. 1999) with several other studies identifying Saharan dust up to
~5km for summertime dust transport (Mortier et al. 2016, Veselovskii et al. 2016, Tesche et al. 2011). This indicates
that the model may be transporting too much dust aerosol and water vapor higher into the atmosphere and this transport
is well correlated. Correlations over North America and Eastern Europe are strongest in the BL to lower FT.
Wintertime correlations over East Asia/Beijing are pretty consistent throughout the column. Negative correlation
regions associated with smoke aerosol, including the Sahel, Southern Africa, and Southeast Asia have the strongest
correlations in the lower FT and largely disappear beyond this point. The shift in correlations with vertical location
are also evident in the histograms (Figure 12) when compared to the full column distribution. This is particularly the
case for the lower and mid free troposphere where the number of grids with correlations greater than 0.5 increase.
Additionally, the shift of the correlations to mostly positive can be seen in the mid and upper free troposphere
histograms.
**3.6 Impact of Hygroscopic Growth on AOD and PW relationships**
The final consideration in this work is hygroscopicity. Although the effects of clouds on the AOD and PW relationship
is also important to understand, this effect cannot be investigated using NAAPS since the model does not account for
the processing of aerosol in cloud droplet, rapid gas-to-particle conversion in cloud droplets or the high RH halo in
the immediate vicinity of clouds. However, this should be considered in follow-on work. While relationships between
AOD and PW have been demonstrated, this signal can be either from co-transport or a confounding relationship
between enhanced PW and RH. The correlation between RH and PW is shown in Figure 13 by season and for the
boundary layer and parts of the free troposphere. The largest spatial variations in the PW and RH correlation occur in
the boundary layer, as anticipated, with strong correlations found over Africa, extending into the Indian Ocean/India,
located further North during JJA and further South during DJF and similar patterns during MAM and SON. Other
regions of high correlation in the boundary layer include off the coast of South America, parts of Australia, and limited





locations in the tropical oceans. Beyond the boundary layer, the overall patterns are generally consistent throughout
the vertical column with strong correlations in the subtropics and tropics (>0.9) with some variations on the extent by
season. In JJA for example, this high correlation region extends further north while in DJF the high correlation region
extends further in the Southern hemisphere. In this highly correlated region, hygroscopic growth is expected to be a
significant driver in AOD and PW relationships when dust is not the dominant aerosol type. RH and PW correlations
are higher over ocean regions than over land in the northern hemisphere, which should be impactful for sea salt aerosol
and PW correlations. The impact of the RH and PW correlations on the AOD and PW relationship are shown in
calculated seasonal relationships between "dry" AOD, which excludes the impact of hygroscopic growth, and PW in
the NAAPS-RA in Figure 14. This figure includes the correlations, the slope of the "dry" AOD and PW relationship,
and the statistically significant different in mean PW for high "dry" AOD events. The removal of hygroscopic growth
from the AOD calculation had the following outcomes on the resulting correlations: 1) the previously positive
correlation was reduced in magnitude 2) the previously negative correlation coefficient became more negative 3) the
sign of the correlation flipped from positive to negative and 4) little to no change in the correlation. Regions such as
the Eastern United States and Europe fall into the first category where positive AOD and PW correlations are found
for all seasons, but the correlation coefficient is greatly reduced. For the Eastern United States, peak correlation
coefficients were in the approximately 0.6-0.7 range with hygroscopic growth and fell below 0.5 without it. This is
especially true in JJA when RH and PW correlations are the strongest. Likewise, positive correlations in Europe are
still present, but weakened. In these cases, hygroscopic growth amplifies an existing positive relationship that is
somewhat weak when evaluating seasonal data by correlation. In regards to the second category, this corresponds to
regions dominated by smoke aerosol that previously exhibited negative AOD and PW relationships, such as Peninsular
Southeast Asia during the MAM months and Insular Southeast Asia during the SON months. Additionally, increases
in negative correlations are found for aerosol transport from Asia across the Pacific Ocean. In these cases, hygroscopic
growth reduces an existing negative relationship between aerosol and water vapor. Ocean regions mostly account for
the third category where the correlation flipped from a weak positive to negative value. Regions that are dominated
by dust, including the Sahara and Arabian Peninsula, fell into the fourth category as there is no hygroscopic growth
for dust in NAAPS.

In regards to the slope of "dry" AOD and PW, the same categories apply with similar spatial patterns relative to the
correlation analysis. Additionally, the same is true when examining the difference in mean PW for high "dry" AOD
cases. In this case, it is found that either: 1) an increase in PW is still statistically significant, but the difference in
mean PW is much less 2) a decrease in PW for high "dry" AOD cases is still statistically significant with a larger
decrease when not considering hygroscopic growth 3) the sign of the difference flipped from an increase in PW to a
decrease or the difference became statistically insignificant and 4) the PW difference did not change much due to dust
dominated conditions. While the modeled differences in PW are statistically significant when excluding hygroscopic
growth, they are small, with peak differences on the order of a few millimeters. The results here indicate that
hygroscopic growth of aerosol plays an important role in the AOD and PW relationship. While PW is still a good
tracer for AOD as shown in this work, it should be kept in mind that there is a difference in water vapor as a tracer for



AOD and for aerosol mass. It is expected that the relationship between "dry" AOD and PW would be a closer
representation of the dry aerosol mass to PW relationship.

**3.7 Discussion through Example Cases at Individual AERONET Sites**

In order to further understand regional differences in observed AOD and PW relationships, individual sites in which
strong AOD and PW relationships were identified and had several years of observational data available were selected
for further analysis. These sites included: 1) Tallahassee, Florida for Southeast US pollution (Figure 15); 2) Beijing,
China for Asian haze and dust (Figure 16); 3) Izana, Canary Islands for Saharan dust (Figure 17); and Alta Floresta,
Brazil for South American biomass burning (Figure 18). For the four identified AERONET sites, the daily-averaged
AOD and PW timeseries are examined for seasons in which correlations were found to be strong. This includes DJF
for the Tallahassee and Beijing sites and JJA for Izana. At these sites, the identified relationships between AOD and
PW were positive. The Alta Floresta site, which exhibited negative AOD and PW relationships in the presented results,
is further examined for the SON biomass burning season. The daily-averaged data are included since this is what was
analyzed in the previous analyses. Additionally, the AERONET data, without any averaging, is further examined for
individual cases from the site-specific timeseries for which peaks in AOD and/or PW were found. NAAPS-RA AOD
and PW fields are also shown for the selected cases (Figures 15-18).
At the Tallahassee site, the predominant aerosol type is ABF/pollution and although AOD values are generally low
during DJF (mean values in the 0.1-0.2 range, Figure 3), strong AOD and PW relationships were found with
correlations of 0.74 in the AERONET dataset and 0.66 in the NAAPS-RA with PW mean differences around 1cm for
high AOD events (Table 1). The daily-averaged AOD and PW timeseries for the 2018-2019 DJF season are shown in
Figure 15a. The timeseries indicates, consistent with the correlation analysis, that the daily-average AOD and PW
generally move together. There are several joint peaks in AOD and PW that occur during the time period and three
selected cases are examined further, including 1/1/2019, 2/7/2019, and 2/17/2019, with these events identified in the
Figure15a timeseries using red arrows. The AERONET AOD and PW timeseries for these three cases are shown in
Figures 15b-d, respectively. The 2/17 case has the least data points available, making it harder to evaluate diurnal
changes in AOD and PW, however, the 1/1 and 2/7 cases have a good number of data points throughout the afternoon
and later into the evening. For these two cases in particular, the changes in AOD and PW throughout the day are
generally consistent with each other, indicating that the AOD and PW relationships can extend to sub-daily timescales.
AOD and PW plots for the three identified cases are shown from the NAAPS-RA in Figure 15e as a means to assess
the types of aerosol events that are impacting Tallahassee when coordinated peaks in AOD and PW are observed. For
all three cases, coincident transport of AOD and PW is observed in the reanalysis fields, associated with a frontal
system. This type of frontal transport was commonly found for events in which coincident PW and AOD peaks are
observed at Tallahassee. As the DJF season in the Southeast United States has significant frontal activity, this is likely
an important factor in enhanced AOD/PW relationships during this season.
Beijing is an urban site that commonly experiences high AOD levels related to pollution as well as transported dust
and smoke events. Wintertime events are notorious for exhibiting some of the worst air quality in the world for a major
population center (e.g., Wang et al., 2014; Gao et al., 2016; Zhang et al., 2018). Additionally, Beijing has the benefit



of a long AERONET data record with measurements dating back to 2001. Like Tallahassee, the AOD and PW
timeseries at Beijing is evaluated for the 2018-2019 DJF season (Figure 16) with strong positive relationships
identified in both the AERONET and NAAPS datasets with correlations of 0.71 and 0.76, respectively, and the change
in AOD with PW exceeding 1cm⁻¹ (Table 1). As indicated in Figure 4, ABF/pollution is the dominant aerosol type
with some dust present with much higher AOD values observed at this location (Figures 3 and 5). In East Asia,
pollution build-up often occurs under stagnant weather conditions where a stable atmosphere leads to limited vertical
mixing (Wang et al. 2014, Li et al. 2019) with the monsoon being an important factor in determining synoptic
conditions. In particular, the East Asian Winter Monsoon (EAWM) has been shown to be a controlling factor in aerosol
concentrations during the winter season (Li et al. 2016; Jeong et al. 2017). With a strong EAWM, reduced aerosol
concentration occur over northern East Asia, including Beijing, due to stronger northerly winds. In weaker EAWM
years, increased aerosol concentrations occur in the north due to weakened winds and more stagnant conditions. The
daily-averaged AERONET AOD and PW timeseries for 2018-2019 DJF is shown in Figure 16a. Consistent with the
previously presented evaluations, the daily-average time series for this particular DJF time period are well correlated
with AOD and PW moving up and down together. As was done for the Tallahassee site, several peaks in AOD and
PW were selected for further evaluation and are highlighted with red arrows on Figure 16a, including the peak on 1/3
and its subsequent decrease on 1/4/19, and the peak on 12/20/18 and its subsequent decrease on 12/21/18. The non-
averaged AERONET AOD and PW timeseries for these two cases are shown in Figure 16b and c, respectively, with
a zoomed in view of 12/21 on Figure 16d. For both of these events, high AOD from ABF/pollution and PW values
are observed with a subsequent dropoff the following day, which are well coordinated in the full dataset. The closer
look at the data on 12/21/18 (16d), like the previous Tallahassee examples, shows consistent movement between the
measured AOD and PW indicating the presence of correlations on short timescales. For both of these events, the
NAAPS-RA AOD and PW fields are shown for both the peak and subsequent dropoff in Figure 16e. The movement
of the large-scale air mass can be seen in both the AOD and PW fields. For the 1/3/19-1/4/19 event, NAVGEM
meteorological fields indicate weakened northerly winds due to a region of high pressure over the eastern portion of
the continent, leading to stagnant conditions at the surface in Beijing and local pollution and water vapor build-up. On
the following day, the high pressure system moved eastward. As a result, the Siberian High northerlies were no longer
suppressed and a more typical wintertime circulation resumes with the winds jointly pushing the aerosol and water
vapor southward and away from Beijing. For the 12/20/18-12/21/18 case, extensive multi-level cloud cover can be
seen in both MODIS Terra and Aqua images on 12/20, indicating that this may be a case where clouds may have
played a role in gas-to-particle conversion in the polluted air and/or enhanced particle humidification in the high RH
fields associated the clouds, consistent with the findings of Eck et al. (2018). As the air mass moves on 12/21, a joint
reduction in both AOD and PW is observed at Beijing, demonstrating the impact of large scale transport. Thus in the
Beijing case, the overall regional weather patterns are an important factor in the AOD and PW relationship.

The third site that is examined is Izana in the Canary Islands (Figure 17). The Izana site, which is located
approximately 300km west of the African coast, is particularly useful for evaluating aerosol and water vapor
relationships for free tropospheric dust. It has also been noted in the community that against a Saharan air layer free





subsidence regime, the infrared signals of Saharan Air Layer dust are quite small relative to co-transported water vapor
(Gutleben et al., 2019; Ryder, 2021: Barreto et al., 2022). Nevertheless, forecasters find the water vapor signal useful
in tracking dust events (Kuciauskas et al., 2018). Izana is a mountain site located at approximately 2400m, above a
strong subtropical temperature inversion layer which makes it ideal for monitoring free tropospheric plumes. In the
summer months, frequent and intense Saharan air mass outbreaks in the subtropical free troposphere impact the site.
Particularly large AOD dust storms were observed transporting Saharan dust across the Atlantic in the summer of
2020, including the so called "Godzilla" dust event in June that has been examined in detail in previous studies (Francis
et al. 2020). As a result, this time period was evaluated in further detail at Izana with a focus on several dust events.
In the analysis conducted in this work, positive relationships between AOD and PW were found in both the AERONET
and NAAPS datasets at Izana during the JJA season with the AERONET dataset having a correlation of 0.66 and the
NAAPS-RA indicating a weaker correlation of 0.48 (Table 3). As noted for other sites, this difference may be due to
altitude effects at the Izana site which may not be captured in NAAPS. The daily-average AOD and PW timeseries
for the 2020 JJA season at Izana are shown in Figure 17a. The AOD and PW are pretty well correlated, although there
are PW peaks present without the presence of aerosol. As sources of aerosol and water vapor are different, this is not
unexpected. Three of the peak from the timeseries as indicated by the red arrows were selected for further evaluation,
including 6/14/20, 7/18/20, and 7/31/20 with timeseries shown in Figures 17b-d, respectively. As was shown for the
previous pollution cases, the AOD and PW are well correlated in the non-averaged dataset showing the presence of
correlations for dust events on timescales less than a day. The NAAPS-RA AOD and PW fields are shown for the
6/14/20 case in addition to the associated dropoff of both AOD and PW on 6/18/20, as well as for the 7/18/20 and
7/31/20 events in Figure 17e. For the 6/14/20 case, CALIPSO indicates dust tops near Izana at around 5km. A cutoff
low was present to the northwest of Africa and a subtropical high over the Western coast of Africa which resulted in
increased dust generation and recirculation. The southwesterly winds transport the dust to Izana on June 14 and at the
same time, moist ocean air shown by the PW fields is transported to Izana as well, resulting in a spike in both AOD
and PW at the AERONET site. On June 18, the moisture and dust begin pushing south and west across the Atlantic,
resulting in decreases in AOD and PW at the same time at Izana. Similar examples of dust and water vapor cotransport
are shown for the 7/18/20 and 7/31/20 cases. These results are consistent with previous studies which have indicated
enhanced water vapor mixing ratios associated with the dust in Saharan Air Layer events (Marsham et al. 2008, Jung
et al. 2013, Kanitz et al. 2014). Thus, Izana is a good example of subtropical free tropospheric co-transport of water
vapor and dust.

The final site evaluated was Alta Floresta, Brazil for the 2019 SON season, when biomass burning is the dominant
aerosol type. Here, there is a strong seasonal dependency: drier seasonal conditions are associated with a July-October
biomass burning season. At this site and other regions of biomass burning, negative correlations between AOD and
PW were identified in the presented evaluations. For Alta Floresta, SON correlations of -0.38 and –0.47 were
generated from the AERONET and NAAPS-RA datasets, respectively (Table 4). The daily-average AOD and PW
timeseries are shown in Figure 18a. An important thing to note about this timeseries is that a clear downward shift in
the PW fields occurs during October, consistent with monthly site climatologies from AERONET (3.61cm in





September, 4.15cm in October, 4.5cm in November). As fires are associated with dry conditions, the AOD fields
decrease as the water vapor increases. This shift towards wetter and decreased aerosol conditions is driving the
seasonal negative correlations for biomass burning regions. However, despite this overall shift, peaks in AOD and PW
are generally positively correlated. Several such cases are highlighted in the timeseries, including 9/15/19, and 9/23/19.
The 9/9/19 event is also highlighted in which the daily-averaged AOD peaks and the PW is at a low. For all three
cases, the AOD and PW timeseries shown in Figures b-d, show good positive correlations with AOD and PW changing
in the same way throughout the day. For the 9/9/19 case, this appears to a more locally driven event with the extent of
the smoke being more limited and the air being drier than the surrounding areas, suggesting a different air mass (Figure
18e). Additionally, there is much more small scale variability in the AOD and PW fields (Figure 18b). For this type
of event, the daily-average PW fields might not be as good of an indicator of what is going on with AOD. For the
other two cases, the NAAPS-RA plots indicate larger spatial extent of the smoke with more moisture associated with
the air mass. In this case, daily-average PW is a better indicator for large scale smoke events. However, for all three
smoke cases, AOD and PW are correlated on an event level. Thus, in this case the AOD-PW relationship identified in
the previously presented AERONET and NAAPS-RA evaluations represents the end of the burning season with wet
season onset in the middle of a "climatological season". However, PW is a good positive indicator of AOD associated
with smoke on an event level, consistent with what has previously been shown in the literature for case study
evaluations.
**4.0 Conclusions and Implications**
The relationship between AOD and PW was evaluated globally and on seasonal and daily timescales using
approximately 20 years of AERONET observational data and the 16-year NAAPS-RA v1.0 model fields. As
AERONET observations have small measurement uncertainties, the observational analysis provides a best estimate
of the AOD and PW relationships. The observational analysis was combined with the NAAPS-RA in order to provide
a complete global perspective on the AOD and PW relationship as well as to provide an avenue for further exploration,
including what the likely drivers of these relationships are, what the relationships look like when taking the vertical
location into account, and the impact of hygroscopic growth on the AOD and PW correlations.
The major findings of this work include:
1.   Seasonal relationships between AOD and PW are present across the globe at both seasonal and daily levels.
Most often AOD and PW relationships are strongly positive at seasonal to daily time scales, especially for
species such as pollution and dust. Biomass burning, however, has negative seasonal relationships due to fire
proclivity in dry seasons. Nevertheless, positive daily relationships are observed, associated with transport.
For regions like the Sahel, negative relationships between AOD and PW were found with spatial patterns
consistent with shifts in the ITCZ in which convection leads to aerosol scavenging.
2.   Mid-latitude relationships between AOD and PW appear to be driven by frontal activity. While
tropical/subtropical relationships are driven by seasonal monsoon activity, ITCZ, and dry season patterns.



Dust transport associated with African easterly waves and cyclones are the link between aerosol and water
vapor for the Sahara.

3.  The observed correlations between the AOD and PW were stronger when evaluated by vertical level with the
strongest correlations identified in the free troposphere, consistent with large scale aerosol and water vapor
transport. The location of the strongest correlations varied by aerosol type with dust dominated regions
having the strongest correlations in the mid free troposphere and smoke dominated regions having the
strongest correlations in the lower free troposphere.

4.  Hygroscopic growth of aerosol particles, which is associated with increased relative humidity and often
occurs with increasing PW, has a large influence on the observed covariability between AOD and PW,
particularly in the mid-latitudes and for non-dust aerosol species. While transport covariance between AOD
and PW is present, the imbedded RH to PW relationship is the dominant term. This indicates that PW is a
good tracer for AOD, but not necessarily aerosol mass.

5.  Covariability between AOD and PW for dust-dominated events is statistically significant and hygroscopic
growth is not an important factor.

Overall, this evaluation provides a global perspective on AOD and PW relationships. As has been previously
shown for individual case studies in the literature, this work reaffirms that PW is a useful tracer for aerosol
transport and such relationships are present across the globe. The seasonal AOD and PW evaluations conducted
in this work highlight regions and seasons for which AOD and PW relationships are expected to be more
prevalent. In particular, regions and seasons for which strong correlations and impacts on the PW distribution for
high AOD events are found are associated with synoptic scale aerosol events, including large-scale pollution and
smoke events over CONUS and Europe, Saharan dust events over the Atlantic, biomass burning events during
regional dry seasons in South/Central America, Africa, and Southeast Asia, and Asian dust/haze events. This is
confirmed when evaluating AOD and PW relationships on an event basis in different parts of the world in which
coincident peaks in daily-averaged AOD and PW were associated with large-scale aerosol transport events. The
vertical evaluation of the AOD and PW relationship provides further evidence that a strong contributor to the
identified relationship is synoptic scale aerosol transport in the free troposphere where the relationships were
found to be stronger than the fully integrated vertical column. These signals were present for all aerosol types
evaluated, indicating PW can be a useful tracer for AOD associated with all aerosol types as long as sources of
both and a common linking transport mechanism are present. For example, in the United States, fronts were the
linking transport mechanism while in East Asia, monsoonal patterns controlled joint transport. Likewise, dust
transport associated with African easterly waves and cyclones linked aerosol and water vapor for the Sahara.
Regions identified with strong correlations indicate the frequent presence of such synoptic scale co-transport
events while the PW distribution evaluation for high AOD events highlights regions in which such events are
present, but can be infrequent, as was the case for Boreal smoke events in summertime.
This work provides a first step in understanding the important aerosol and water vapor relationship on a global
scale. While aerosol and water vapor relationships will vary from air mass to air mass, this analysis provides an



understanding of where and when AOD and PW relationships are expected to be of importance and can be
exploited in 1) using water vapor as an aerosol tracer 2) in data assimilation applications and 3) for radiative
transfer studies in which collocated aerosol and water vapor can impact results. While this analysis provides a
quantitative estimate of the aerosol and water vapor relationship in a big picture sense, the next step is to further
understand the aerosol and water vapor relationships on an event level. This is particularly important for data
assimilation in which an understanding of how this relationship temporally and spatially evolves for individual
air masses needs to be developed. As such, a follow-on study will be conducted to investigate the evolution of
aerosol and water vapor in space and time on an event level with a focus on specific regions identified in this
work.
**Code and data availability**
AERONET observations are available for download through https://aeronet.gsfc.nasa.gov/ and the NAAPS
reanalysis data in NetCDF format can be downloaded through The US Global Ocean Data Assimilation
Experiment (GODAE) server (https://usgodae.org/).
**Author contributions**
Authors Juli Rubin and Jeffrey Reid planned the analysis while Juli Rubin conducted the majority of the analyses
presented in this work. Peng Xian provided the NAAPS-RA dataset and provided help in using the data. Jeffrey
Reid, Juli Rubin, Christopher Selman and Thomas Eck helped in interpreting the results of the study with Chris
Selman focused on the meteorological aspect and Thomas Eck providing important feedback on the evaluation
using AERONET data.
**Competing interests**
The peer-review process was guided by an independent editor, and the authors have no other competing interests
to declare.





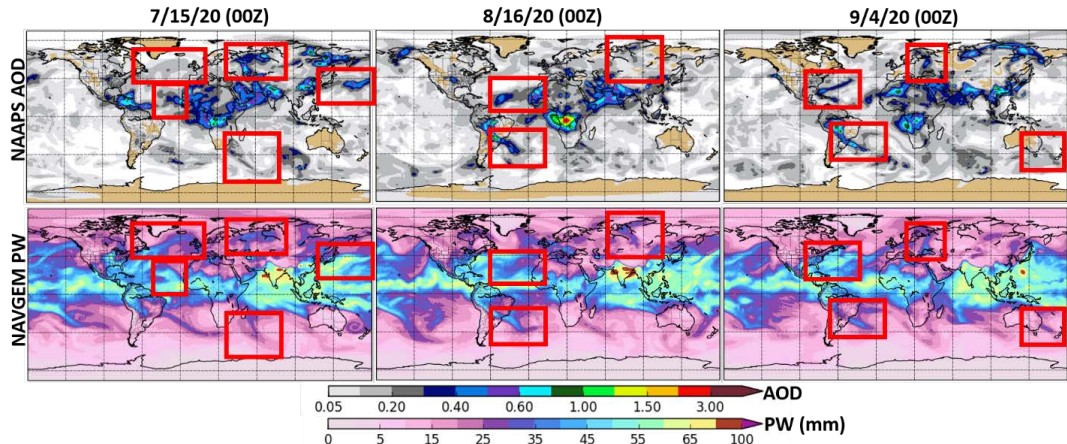


**Figure 1. Examples of NAAPS AOD and NAVGEM PW forecasts in which similar synoptic scale transport patterns are found, particularly in the mid-latitudes. Aerosol and water vapor features with similar transport patterns are highlighted in matching red boxes in the AOD and PW plots. These types of co-transport events of both positive and negative correlation are found in forecasts on a daily basis.**

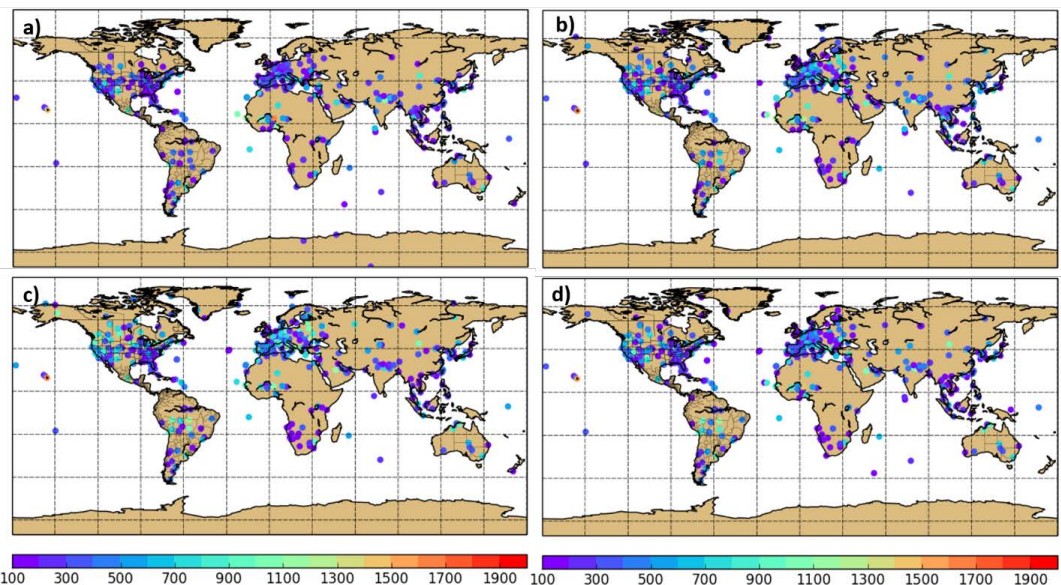

831

**Figure 2. Count of daily-averaged AERONET AOD and PW data points by season: a) DJF b) MAM c) JJA d) SON. Only sites with at least 100 points are shown.**

834

835



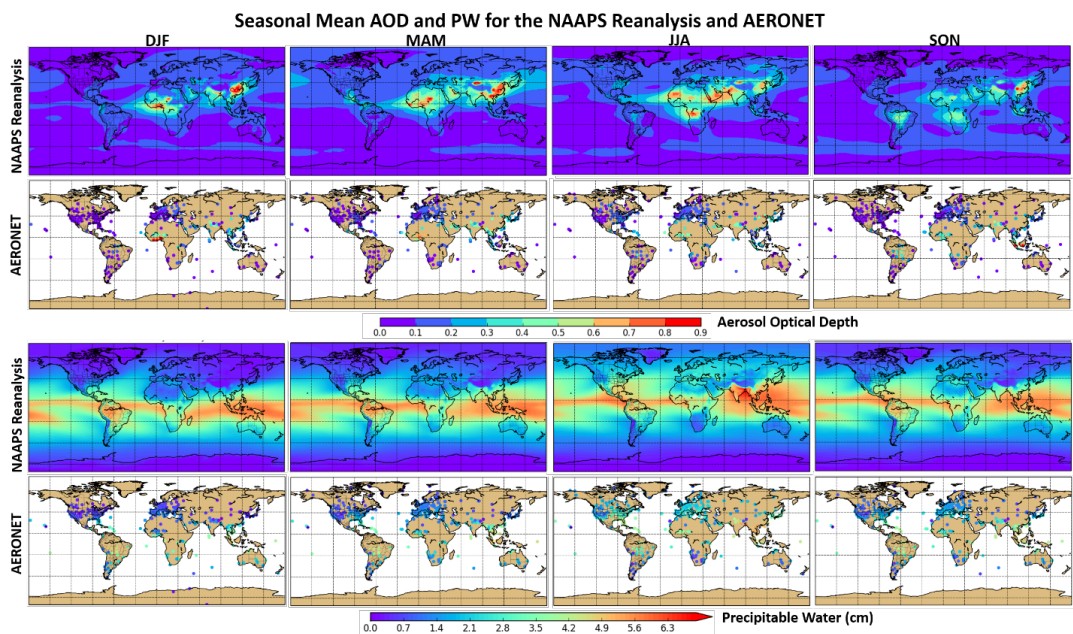

**Figure 3. Mean AOD and Precipitable Water (cm) for the NAAPS-RA and at AERONET sites by season: DJF, MAM, JJA, SON.**

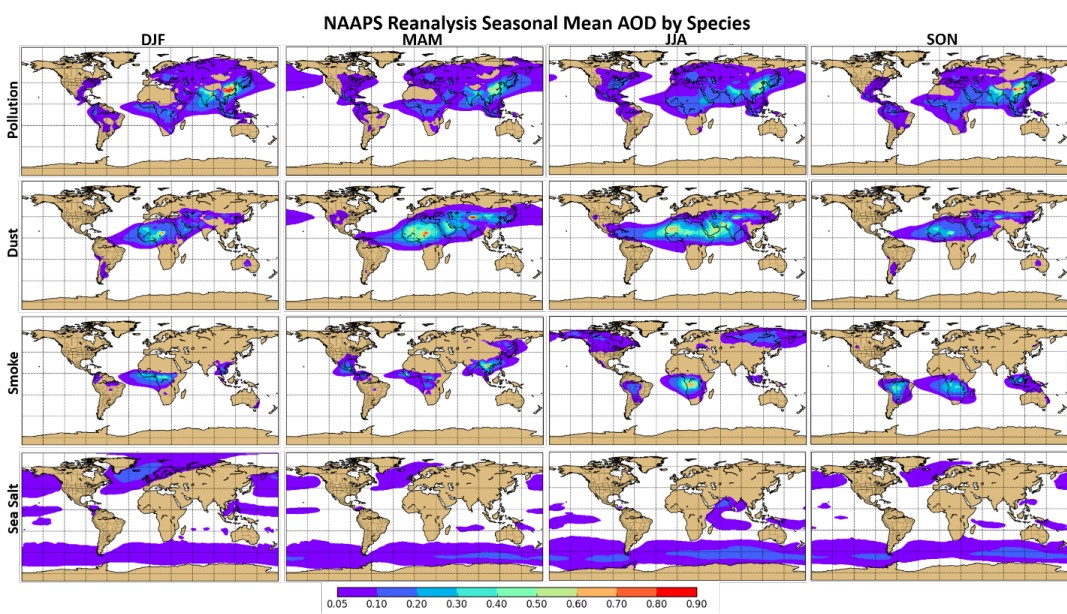

**Figure 4. NAAPS-RA seasonally averaged AOD by aerosol type, including pollution (anthropogenic and biogenic fine aerosol), dust, smoke, and sea salt.**

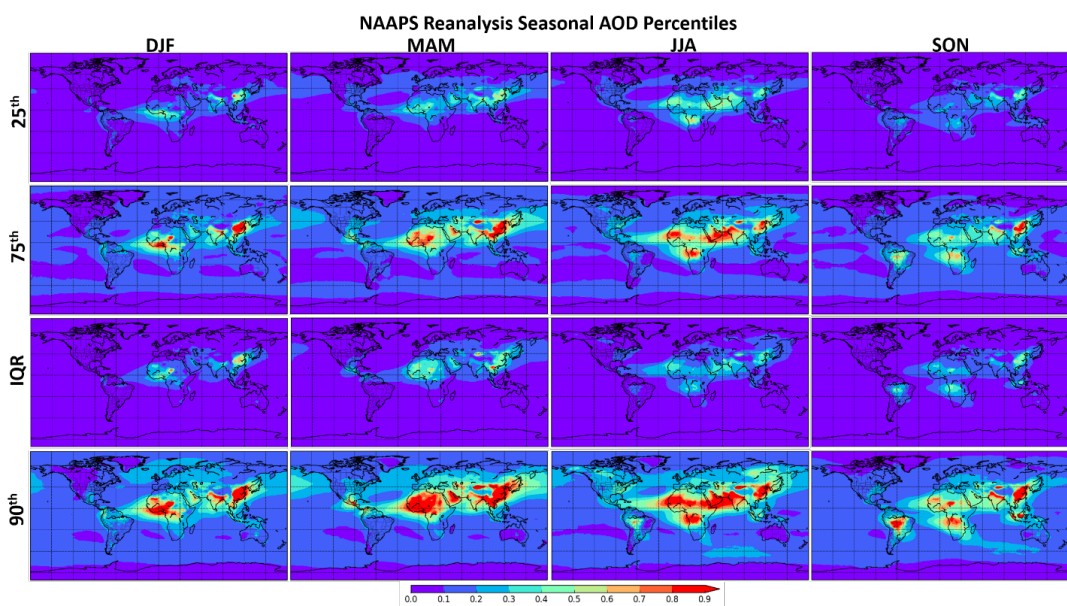


**Figure 5. NAAPS-RA AOD percentiles by season (DJF, MAM, JJA, SON). The 25th and 75th percentiles are shown along with the interquartile range (IQR). The 90th percentile is used to show high AOD values at a given location.**



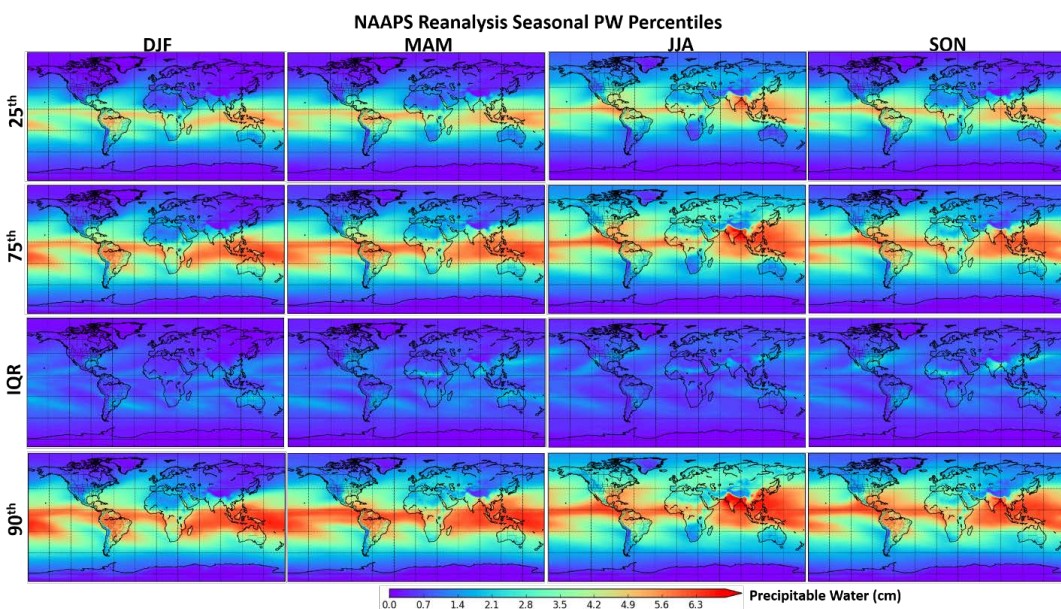


**Figure 6. NAAPS-RA PW percentiles by season (DJF, MAM, JJA, SON). The 25th and 75th percentiles are shown along with the interquartile range (IQR). The 90th percentile is used to show high PW values at a given location.**




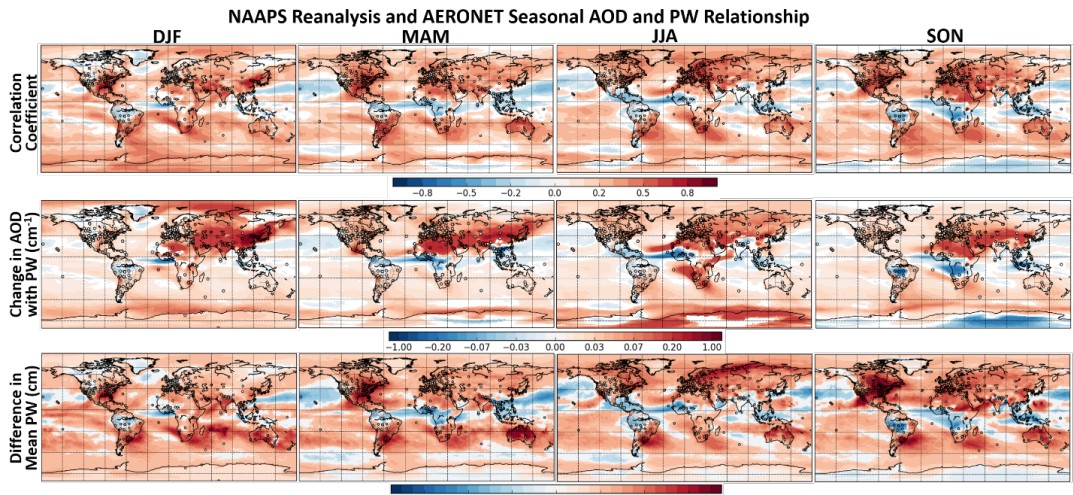


**Figure 7. Seasonal AOD and PW relationships based on AERONET data (circles) and the NAAPS-RA (global map) shown as: a) correlation coefficients between daily-averaged AOD and PW (non-zero values are statistically significant at the 95% level) b) Theil-Sen regression slopes (change in AOD with PW) between daily-averaged AOD and PW in units of cm-1 at locations where the correlation is statistically significant and c) the statistically significant difference in mean PW (cm) between the PW distribution associated with high AOD events (> 1 standard deviation above mean) and the PW distribution for all AOD values. Red regions indicate a positive relationship between AOD and PW (higher moisture conditions for higher AOD events) and blue regions indicate a negative relationship (dryer conditions for higher AOD events).**

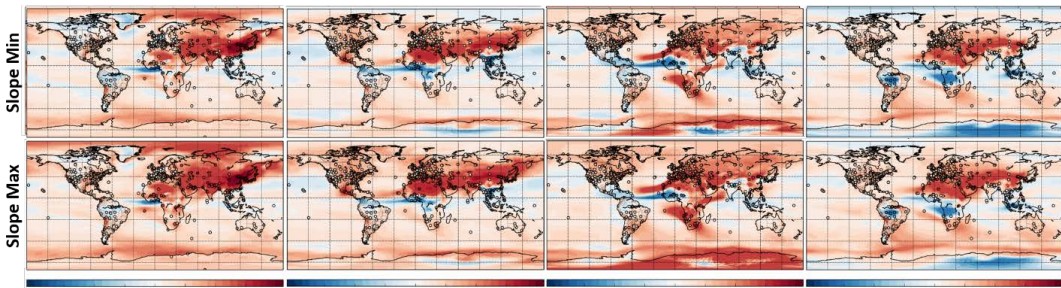


**Figure 8. The 95% confidence interval in the Theil-Sen change in AOD with PW (cm-1) for DJF, MAM, JJA, and SON. The confidence intervals are shown for both the NAAPS-RA and AERONET.**






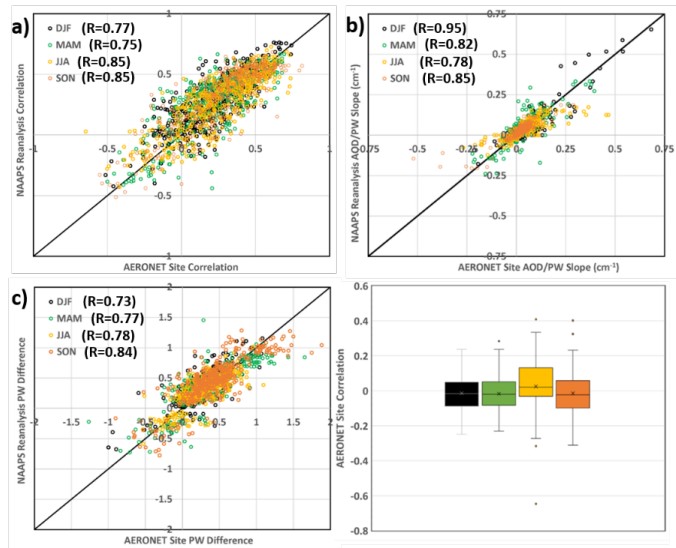


**Figure 9. Scatterplot comparisons of the NAAPS-RA and AERONET: a) AOD and PW correlations at AERONET sites b)**
**the change in AOD with PW (cm-1) at AERONET sites and c) the statistically significant difference in mean PW associated**
**with high AOD events compared to the full PW distribution at AERONET sites. The comparisons are shown by season**
**(DJF, MAM, SON, JJA) and correlations between the datasets are included. Additionally, the distribution of AERONET**
**site correlations for which sign differences were found between NAAPS and AERONET calculated AOD/PW relationships**
**are shown seasonally in d).**

870



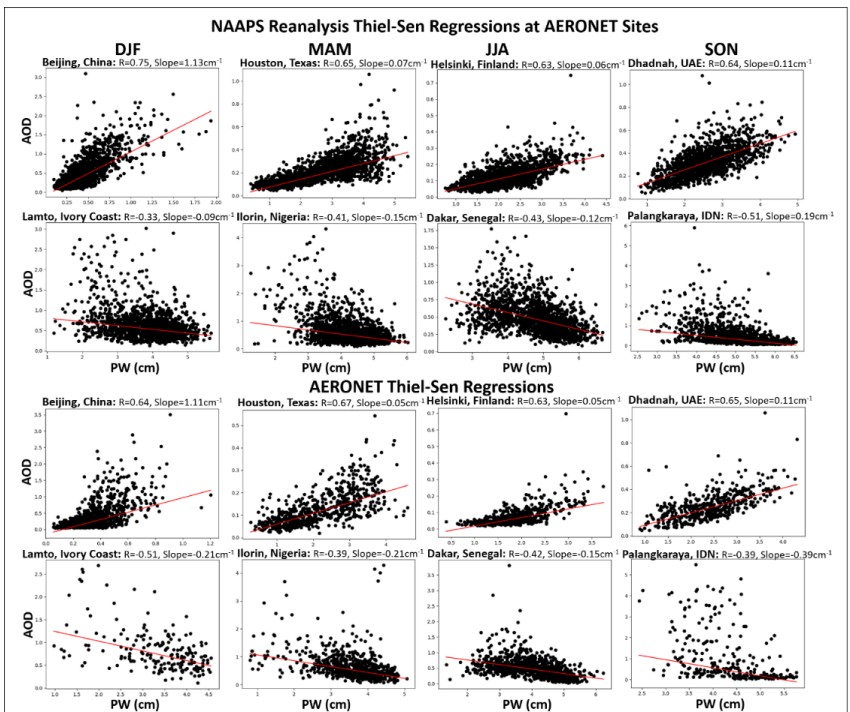

871

**Figure 10. Seasonal examples of NAAPS-RA and AERONET Theil-Sen regression calculations for positive correlation locations (Beijing, Houston, Helsinki, Dhadnah) and a negative correlation locations (Lamto, Ilorin, Dakar, Palangkaraya). The black dots are the AOD and PW pairs from the NAAPS-RA or AERONET and the red line is the Theil-Sen fitting, which is the median of the slopes for the range of data pairings. The location name, correlation coefficient (R), and the Theil-Sen slope (Slope) are included with each plot.**



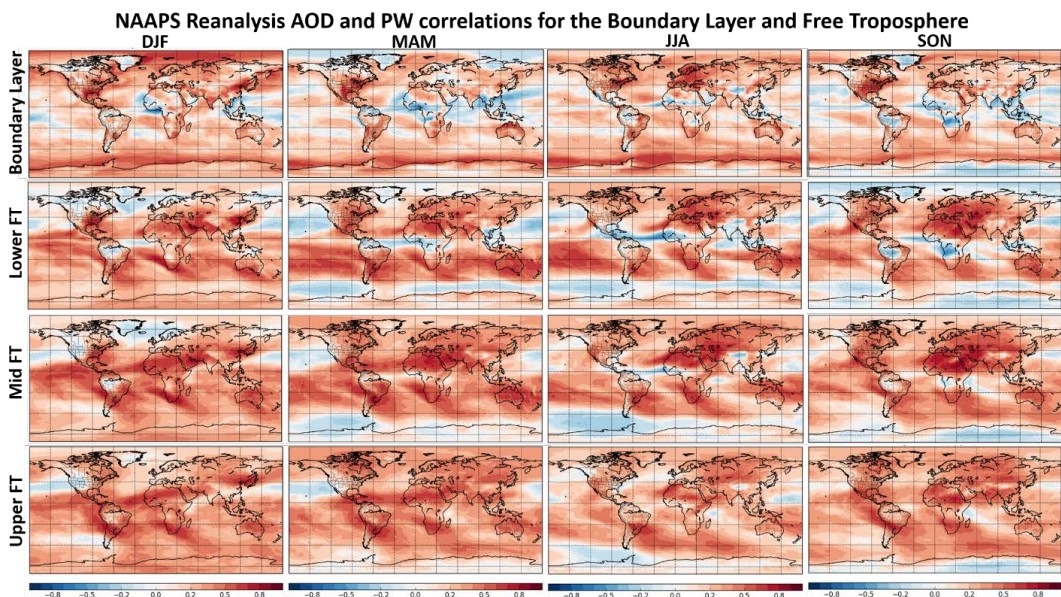


**Figure 11. NAAPS-RA seasonal correlations (DJF, MAM, JJA, SON) between vertically-integrated total aerosol extinction and specific humidity in the boundary layer, lower free troposphere, mid free troposphere and upper free troposphere. Red values indicate a positive correlation and blue values indicate a negative correlation.**






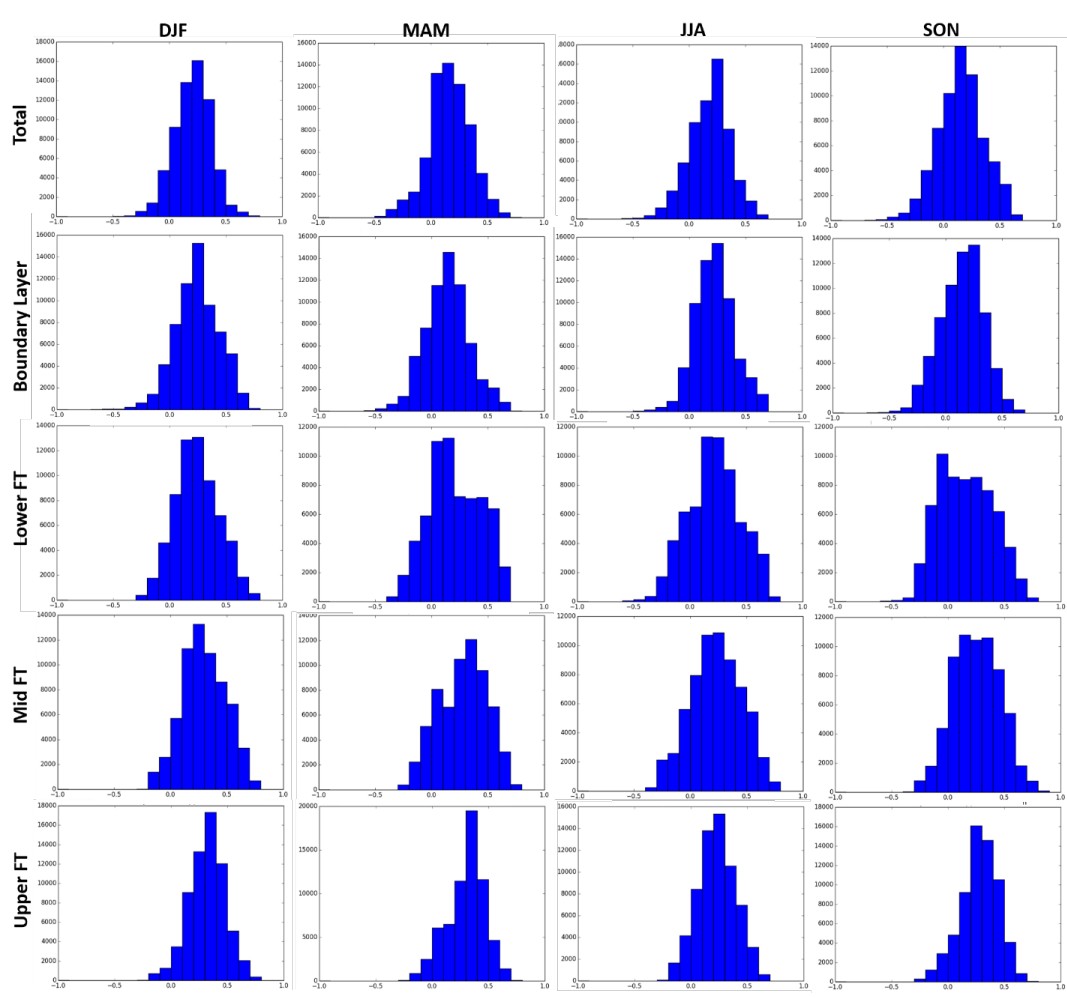

**Figure 12. AOD and PW correlation histograms by season for the full integrated column (Total) and vertical components of the atmosphere (Boundary Layer, Lower/Mid/Upper Free Troposphere).**



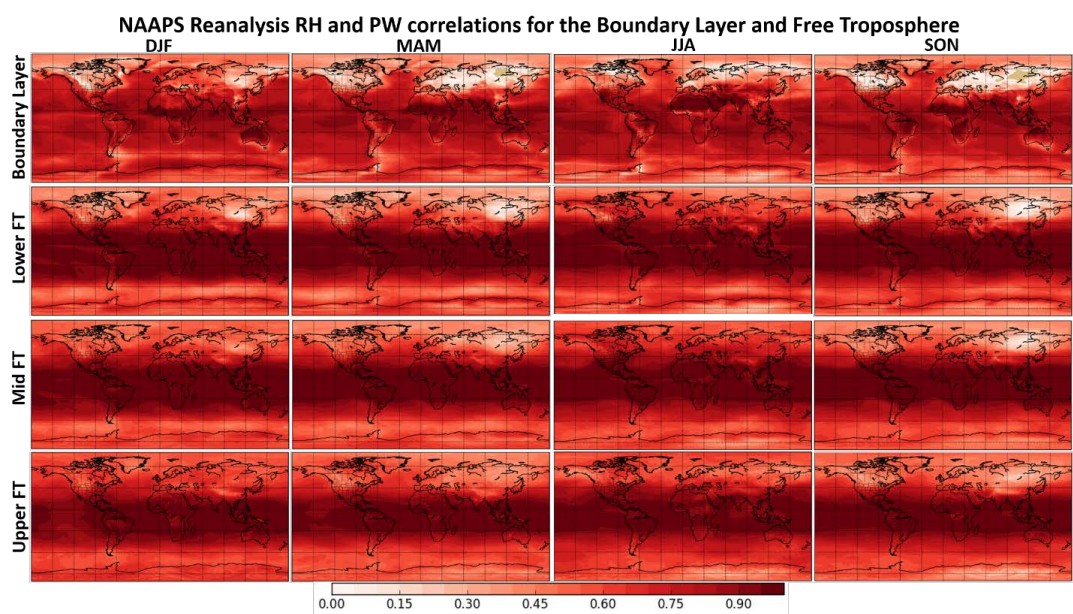


**Figure 13. NAAPS-RA seasonal correlations (DJF, MAM, JJA, SON) between vertically-integrated relative humidity (integrated specific humidity/integrated saturation specific humidity) and specific humidity in the boundary layer, lower free troposphere, mid free troposphere and upper free troposphere.**


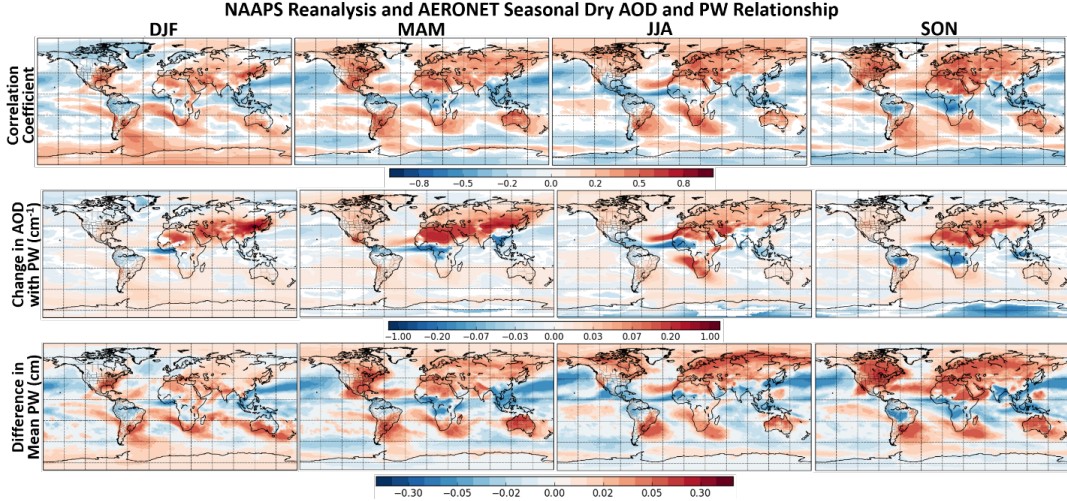


**Figure 14. Seasonal dry AOD and PW relationships based on the NAAPS-RA shown as: a) correlation coefficients between daily-averaged dry AOD and PW (non-zero values are statistically significant at the 95% level) b) Theil-Sen regression slopes (change in AOD with PW) between daily-averaged dry AOD and PW in units of cm-1 at locations where the correlation is statistically significant and c) the statistically significant difference in mean PW (cm) between the PW distribution associated with high dry AOD events (> 1 standard deviation above mean) and the PW distribution for all AOD**





values. Red regions indicate a positive relationship between dry AOD and PW and blue regions indicate a negative
relationship.

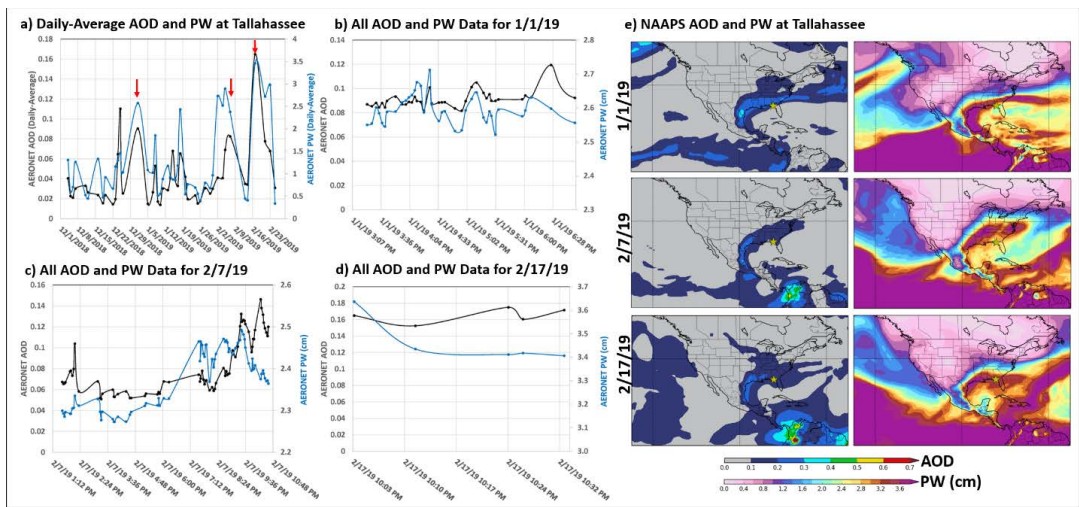


**Figure 15. AOD and PW timeseries at the Tallahassee, Florida site in which strong positive correlations are observed during**
**the DJF season. The daily-average AOD and PW timeseries are shown for the 2018-2019 DJF season with red arrows**
**indicating select events for which joint peaks in AOD and PW are observed (a). Timeseries of AERONET data (non-**
**averaged, all data) for dates identified with red arrows are shown in timeseries b-d. Additionally, NAAPS-RA AOD and**
**PW (cm) fields are shown for the same dates with the AERONET site marked with a yellow star (e).**

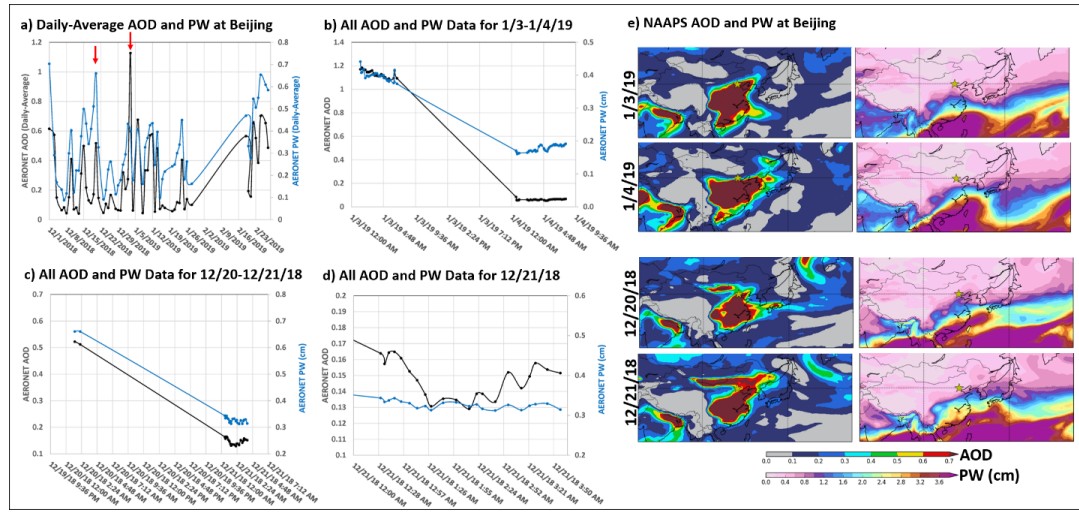


**Figure 16. AOD and PW timeseries at the Beijing, China site in which strong positive correlations are observed during the**
**DJF season. The daily-average AOD and PW timeseries are shown for the 2018-2019 DJF season with red arrows indicating**
**select events for which joint peaks in AOD and PW are observed (a). Timeseries of AERONET data (non-averaged, all**
**data) for dates identified with red arrows are shown in timeseries b-d. Additionally, NAAPS-RA AOD and PW (cm) fields**
**are shown for the same dates with the AERONET site marked with a yellow star (e), including the air mass movement for**
**1/3-1/4/19 and 12/20-12/21/18.**

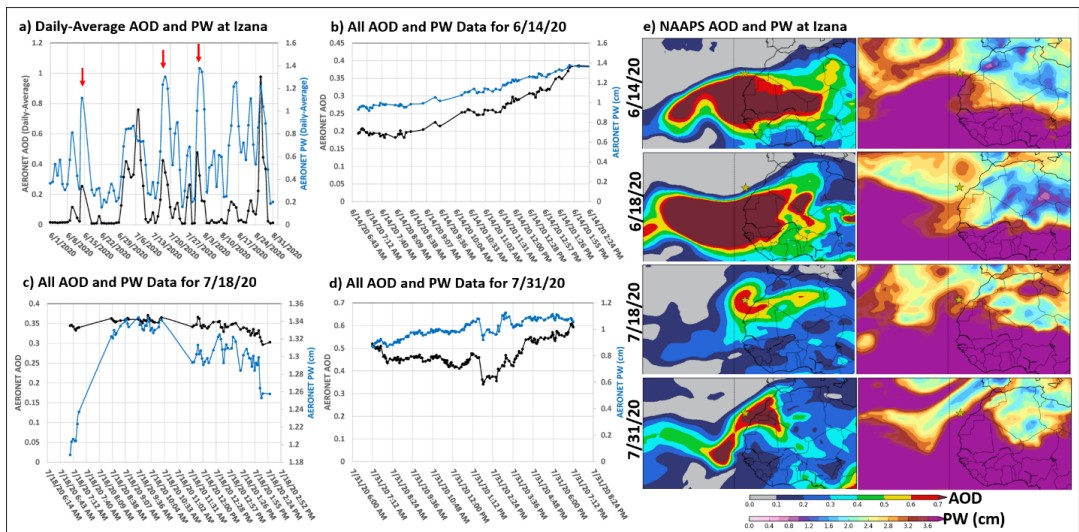


**Figure 17. AOD and PW timeseries at the Izana, Canary Islands site in which strong positive correlations are observed during the JJA season. The daily-average AOD and PW timeseries are shown for the 2020 JJA season with red arrows indicating select events for which joint peaks in AOD and PW are observed (a). Timeseries of AERONET data (non-averaged, all data) for dates identified with red arrows are shown in timeseries b-d. Additionally, NAAPS-RA AOD and PW (cm) fields are shown for the same dates with the AERONET site marked with a yellow star (e). The fields for 6/18/20 are also included which show the joint dip in AOD and PW in the (a) timeseries after the 6/14 event.**

919

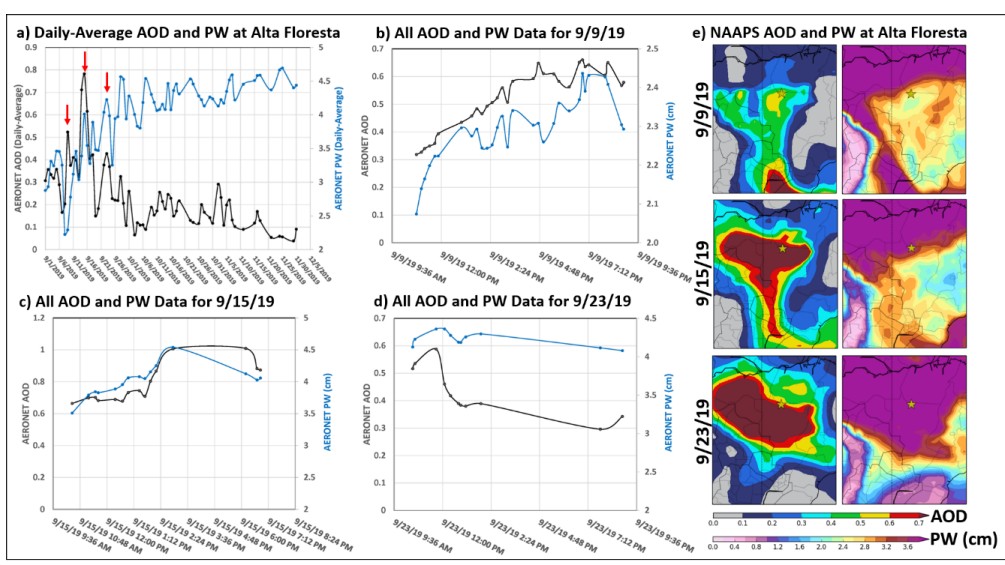

920

**Figure 18. AOD and PW timeseries at the Alta Floresta site in Brazil in which negative AOD and PW correlations were identified in the seasonal analysis. The daily-average AOD and PW timeseries are shown for the 2019 SON season with red arrows indicating select events for further evaluation (a). AERONET AOD and PW timeseries (non-averaged, all data) for the selected events are shown in timeseries b-d. Additionally, NAAPS-RA AOD and PW (cm) fields are shown for the same dates with the AERONET site marked with a yellow star (e).**





**Table 1. AOD and PW relationship evaluation results for DJF at select AERONET sites that exhibited the strongest**
**AERONET correlations, positive and negative. The site name and latitude/longitude information are shown as well as the**
**correlation (R), change in AOD with PW (Slope), and the statistically significant difference in mean PW for high AOD**
**events (PW Diff) for both AERONET and the NAAPS-RA.**

| AERONET Site | Lat | Lon | AERONET | | | NAAPS Reanalysis | | |
|---|---|---|---|---|---|---|---|---|
| | | | R | Slope (cm$^{-1}$) | PW Diff (cm) | R | Slope (cm$^{-1}$) | PW Diff (cm) |
| Stennis | 30.37 | -89.62 | 0.743 | 0.044 | 1.082 | 0.739 | 0.042 | 1.156 |
| Tallahassee | 30.45 | -84.30 | 0.739 | 0.024 | 1.068 | 0.664 | 0.037 | 0.901 |
| Beijing-CAMS | 39.93 | 116.32 | 0.707 | 1.016 | 0.225 | 0.755 | 1.132 | 0.347 |
| XiangHe | 39.75 | 116.96 | 0.673 | 1.349 | 0.196 | 0.755 | 1.132 | 0.347 |
| SEARCH-Centreville | 32.90 | -87.25 | 0.665 | 0.028 | 0.931 | 0.706 | 0.045 | 1.000 |
| Univ_of_Houston | 29.72 | -95.34 | 0.661 | 0.039 | 0.734 | 0.762 | 0.058 | 1.109 |
| SEARCH-OLF | 30.55 | -87.38 | 0.659 | 0.025 | 0.820 | 0.720 | 0.039 | 1.069 |
| Beijing | 39.98 | 116.38 | 0.638 | 1.108 | 0.211 | 0.755 | 1.132 | 0.347 |
| UAHuntsville | 34.73 | -86.64 | 0.633 | 0.029 | 0.930 | 0.684 | 0.049 | 0.866 |
| UH_Coastal_Center | 29.39 | -95.04 | 0.624 | 0.039 | 0.862 | 0.762 | 0.058 | 1.109 |
| Dhadnah | 25.51 | 56.32 | 0.610 | 0.120 | 0.535 | 0.548 | 0.110 | 0.653 |
| Ilorin | 8.48 | 4.67 | -0.261 | -0.100 | -0.513 | -0.159 | -0.063 | -0.356 |
| Pontianak | 0.08 | 109.19 | -0.345 | -0.054 | -0.476 | -0.255 | -0.029 | -0.278 |
| Koforidua_ANUC | 6.11 | -0.30 | -0.459 | -0.194 | -0.894 | -0.424 | -0.178 | -0.676 |
| Jambi | -1.63 | 103.64 | -0.473 | -0.110 | -0.942 | -0.217 | -0.039 | -0.392 |
| LAMTO-STATION | 6.22 | -5.03 | -0.513 | -0.214 | -0.999 | -0.331 | -0.093 | -0.644 |


**Table 2. AOD and PW relationship evaluation results for MAM at select AERONET sites that exhibited the strongest**
**AERONET correlations, positive and negative. The site name and latitude/longitude information are shown as well as the**
**correlation (R), change in AOD with PW (Slope), and the statistically significant difference in mean PW for high AOD**
**events (PW Diff) for both AERONET and the NAAPS-RA.**

| AERONET Site | Lat | Lon | AERONET | | | NAAPS Reanalysis | | |
|---|---|---|---|---|---|---|---|---|
| | | | R | Slope (cm$^{-1}$) | PW Diff (cm) | R | Slope (cm$^{-1}$) | PW Diff (cm) |
| UH_Coastal_Center | 29.39 | -95.04 | 0.694 | 0.041 | 1.099 | 0.665 | 0.068 | 1.058 |
| UMBC | 39.25 | -76.71 | 0.680 | 0.042 | 1.219 | 0.602 | 0.053 | 0.870 |
| Stennis | 30.37 | -89.62 | 0.680 | 0.046 | 0.967 | 0.630 | 0.048 | 0.978 |
| NEON_OSBS | 29.69 | -81.99 | 0.679 | 0.032 | 1.241 | 0.511 | 0.038 | 0.737 |
| NEON_TALL | 32.95 | -87.39 | 0.677 | 0.032 | 0.915 | 0.604 | 0.047 | 0.815 |
| Univ_of_Houston | 29.72 | -95.34 | 0.666 | 0.048 | 1.000 | 0.665 | 0.068 | 1.058 |
| UAHuntsville | 34.73 | -86.64 | 0.664 | 0.049 | 0.992 | 0.611 | 0.050 | 0.766 |
| CCNY | 40.82 | -73.95 | 0.663 | 0.056 | 1.132 | 0.614 | 0.056 | 0.889 |
| CASLEO | -31.80 | -69.30 | 0.652 | 0.020 | 0.342 | 0.432 | 0.044 | 0.218 |
| NEON_ORNL | 35.96 | -84.28 | 0.649 | 0.033 | 1.007 | 0.586 | 0.051 | 0.698 |
| Nainital | 29.36 | 79.46 | 0.629 | 0.256 | 0.422 | 0.388 | 0.112 | 0.457 |
| Midway_Island | 28.21 | -177.38 | -0.326 | -0.033 | -0.370 | -0.323 | -0.026 | -0.490 |
| Mandalay_MTU | 21.97 | 96.19 | -0.338 | -0.049 | -0.439 | -0.384 | -0.080 | -0.717 |
| LAMTO-STATION | 6.22 | -5.03 | -0.353 | -0.177 | -0.429 | -0.321 | -0.132 | -0.350 |
| Vientiane | 17.99 | 102.57 | -0.355 | -0.130 | -0.451 | -0.377 | -0.144 | -0.408 |
| Chiang_Mai_Met_St | 18.77 | 98.97 | -0.372 | -0.109 | -0.586 | -0.391 | -0.126 | -0.554 |
| Jambi | -1.63 | 103.64 | -0.392 | -0.034 | -1.265 | -0.206 | -0.020 | -0.220 |
| Ilorin | 8.48 | 4.67 | -0.392 | -0.210 | -0.710 | -0.412 | -0.151 | -0.726 |
| NGHIA_DO | 21.05 | 105.80 | -0.455 | -0.216 | -0.632 | -0.318 | -0.166 | -0.349 |
| Djougou | 9.76 | 1.60 | -0.500 | -0.222 | -0.836 | -0.385 | -0.114 | -0.700 |








Table 3. AOD and PW relationship evaluation results for JJA at select AERONET sites that exhibited the strongest AERONET correlations, positive and negative. The site name and latitude/longitude information are shown as well as the correlation (R), change in AOD with PW (Slope), and the statistically significant difference in mean PW for high AOD events (PW Diff) for both AERONET and the NAAPS-RA. PW difference values of 0 in the NAAPS-RA indicate the change was not statistically significant.

| AERONET Site | Lat | Lon | AERONET | | | NAAPS Reanalysis | | |
|---|---|---|---|---|---|---|---|---|
| | | | R | Slope (cm$^{-1}$) | PW Diff (cm) | R | Slope (cm$^{-1}$) | PW Diff (cm) |
| Huambo | -12.87 | 15.70 | 0.737 | 0.365 | 0.447 | 0.467 | 0.174 | 0.350 |
| DRAGON_OLNES | 39.15 | -77.07 | 0.731 | 0.088 | 0.894 | 0.399 | 0.048 | 0.386 |
| Pretoria_CSIR-DPSS | -25.76 | 28.28 | 0.731 | 0.140 | 0.438 | 0.660 | 0.120 | 0.446 |
| DRAGON_CLLGP | 38.99 | -76.91 | 0.719 | 0.085 | 1.097 | 0.348 | 0.039 | 0.387 |
| Durban_UKZN | -29.82 | 30.94 | 0.711 | 0.111 | 0.670 | 0.687 | 0.131 | 0.567 |
| Raciborz | 50.08 | 18.19 | 0.672 | 0.065 | 0.749 | 0.614 | 0.085 | 0.578 |
| Izana | 28.31 | -16.50 | 0.662 | 0.240 | 0.345 | 0.477 | 0.152 | 0.495 |
| Helsinki | 60.20 | 24.96 | 0.633 | 0.052 | 0.790 | 0.631 | 0.060 | 0.698 |
| IMS-METU-ERDEML | 36.57 | 34.26 | 0.632 | 0.099 | 0.543 | 0.559 | 0.073 | 0.484 |
| CLUJ_UBB | 46.77 | 23.55 | 0.632 | 0.091 | 0.529 | 0.602 | 0.100 | 0.438 |
| Pokhara | 28.19 | 83.98 | -0.349 | -0.111 | -0.452 | 0.029 | 0.026 | 0.000 |
| Bidi_Bahn | 14.06 | -2.45 | -0.360 | -0.175 | -0.244 | -0.251 | -0.060 | -0.305 |
| Ouagadougou | 12.42 | -1.49 | -0.385 | -0.180 | -0.292 | -0.205 | -0.059 | -0.211 |
| Lumbini | 27.49 | 83.28 | -0.388 | -0.157 | -0.434 | -0.098 | -0.013 | -0.171 |
| Dakar | 14.39 | -16.96 | -0.418 | -0.146 | -0.528 | -0.428 | -0.122 | -0.584 |
| Agoufou | 15.35 | -1.48 | -0.491 | -0.209 | -0.533 | -0.258 | -0.060 | -0.346 |
| IER_Cinzana | 13.28 | -5.93 | -0.501 | -0.269 | -0.424 | -0.279 | -0.077 | -0.259 |
| Jomsom | 28.78 | 83.71 | -0.646 | -0.064 | -0.581 | 0.029 | 0.026 | 0.000 |

Table 4. AOD and PW relationship evaluation results for SON at select AERONET sites that exhibited the strongest AERONET correlations, positive and negative. The site name and latitude/longitude information are shown as well as the correlation (R), change in AOD with PW (Slope), and the statistically significant difference in mean PW for high AOD events (PW Diff) for both AERONET and the NAAPS-RA. PW difference values of 0 in the NAAPS-RA indicate the change was not statistically significant.

| AERONET Site | Lat | Lon | AERONET | | | NAAPS Reanalysis | | |
|---|---|---|---|---|---|---|---|---|
| | | | R | Slope (cm$^{-1}$) | PW Diff (cm) | R | Slope (cm$^{-1}$) | PW Diff (cm) |
| USDA | 39.03 | -76.88 | 0.815 | 0.088 | 1.883 | 0.570 | 0.036 | 1.043 |
| SEARCH-Centreville | 32.90 | -87.25 | 0.796 | 0.026 | 1.748 | 0.531 | 0.032 | 0.917 |
| St_Louis_University | 38.64 | -90.23 | 0.724 | 0.032 | 1.369 | 0.574 | 0.037 | 1.188 |
| Martova | 49.94 | 36.95 | 0.706 | 0.070 | 0.590 | 0.531 | 0.064 | 0.633 |
| UMBC | 39.25 | -76.71 | 0.699 | 0.035 | 1.473 | 0.570 | 0.036 | 1.043 |
| Poprad-Ganovce | 49.04 | 20.32 | 0.689 | 0.053 | 0.651 | 0.568 | 0.071 | 0.525 |
| NEON_TALL | 32.95 | -87.39 | 0.679 | 0.025 | 1.184 | 0.531 | 0.032 | 0.917 |
| GISS | 40.80 | -73.96 | 0.673 | 0.062 | 1.369 | 0.556 | 0.033 | 0.950 |
| Harvard_Forest | 42.53 | -72.19 | 0.666 | 0.035 | 1.051 | 0.576 | 0.036 | 1.020 |
| Mingo | 36.97 | -90.14 | 0.655 | 0.040 | 1.500 | 0.594 | 0.038 | 1.186 |
| Dhadnah | 25.51 | 56.32 | 0.651 | 0.106 | 0.757 | 0.643 | 0.115 | 0.706 |
| Midway_Island | 28.21 | -177.38 | -0.344 | -0.013 | -0.431 | -0.172 | -0.007 | -0.339 |
| Alta_Floresta | -9.87 | -56.10 | -0.382 | -0.148 | -0.573 | -0.470 | -0.197 | -0.634 |
| Koforidua_ANUC | 6.11 | -0.30 | -0.384 | -0.161 | -0.173 | -0.252 | -0.046 | -0.330 |
| Rio_Branco | -9.96 | -67.87 | -0.426 | -0.121 | -0.496 | -0.430 | -0.145 | -0.643 |
| Abracos_Hill | -10.76 | -62.36 | -0.430 | -0.226 | -0.351 | -0.401 | -0.205 | -0.373 |
| Ilorin | 8.48 | 4.67 | -0.431 | -0.091 | -0.653 | -0.227 | -0.033 | -0.476 |
| Palangkaraya | -2.23 | 113.95 | -0.443 | -0.385 | -0.549 | -0.512 | -0.195 | -0.886 |
| Ji_Parana_SE | -10.93 | -61.85 | -0.483 | -0.197 | -0.659 | -0.412 | -0.203 | -0.393 |
| Pontianak | 0.08 | 109.19 | -0.536 | -0.521 | -0.644 | -0.400 | -0.145 | -0.454 |
| Kuching | 1.49 | 110.35 | -0.552 | -0.373 | -0.528 | -0.329 | -0.111 | -0.360 |



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
