# Peer review of "A Global Evaluation of Daily to Seasonal Aerosol and Water"

_Atmospheric Chemistry and Physics, 2022_

## Referee Comment (RC1)

**A Global Evaluation of Daily to Seasonal Aerosol and Water Vapor Relationships Using a Combination of AERONET and NAAPS Reanalysis Data**

Juli I. Rubin, Jeffrey S. Reid, Peng Xian, Christopher M. Selman, and Thomas F. Eck
**acp-2022-594**

The paper comprehensively examines the relationships between column-integrated Aerosol Optical Depth (AOD) and column-integrated precipitable water (PW) over the globe on seasonal and daily timescales using ground-based AERONET sun photometer measurements and NAAPS-RA model fields.

Primary findings reported include:
- Positive relationships are common and significant on daily and seasonal scales, often for pollution and dust
- Negative relationships are less common, but are robust features when they occur such as associated with biomass burning events
- Synoptic meteorology such as mid latitude frontal systems, large anticyclones, ITCZ and monsoon circulations are responsible for large scale patterns of positive relationships between AOD and PW
- Stronger AOD and PW relationships were observed in the free troposphere and found to be consistent with large-scale transport processes
- Hygroscopic growth of aerosols size related to increased AOD was further found to be consistent with broad areas of high PW, and was noted to have a large influence on observed co-variability between the two observables
- For dust dominated episodes especially in arid air masses, the co-variability between AOD and PW is significant – but hygroscopic growth does not appear to be a factor
- These findings broadly affirmed conditions by which the co-variance with PW and AOD signals are robust and allow PW to be a predictor of AOD transport

**Overall Assessment: Publish with minor technical corrections**

**Strengths:**
The study is built upon the long, well-characterized and extensive AERONET ground-based measurement record (> 20 years) with the NAAPS-RA state-of-the-art model system that incorporates MODIS and MISR AOD satellite records (>15 years) with the full array of conventional global meteorological temperature, humidity, and wind observations through the Navy's operational forecasting system.

The references are comprehensive and provide a fair assessment of the state of knowledge on the measurement systems and related processes that influence AOD and PW relationships.

The approach is sound and based upon statistical analyses that are straightforward and easy to comprehend, and thereby, extend insight gained from each observational record. The study

further seeks to understand regional and seasonal scale features, yet, examined shorter term fluctuations to ensure that relationships identified between AOD and PW tracked each other as postulated for the larger variations.

The interpretation of the statistical findings are grounded in knowledge of aerosol sources, instrument performance, and model capabilities and do not reach beyond confidence estimates provided by statistical metrics.

The paper is lucid and flows well. It provides a healthy balance between details on the study approach and findings on AOD and PW relationships.

**Weakness:**

The figures are very small and details are difficult to read in print versions. When viewed on a screen with zoom capabilities, the figures are mostly legible and highlight 2d structures as described in the text. The labels, however, need to be revised to be readable in a print version – unless print versions are no longer a priority.

A few minor text changes are suggested below.
- Introduce the definition of ABF on page 6
- Clarify the timeframe for appearance of the negative correlation in spring/summer months (page 11). Is this boreal spring/summer?

---

## Referee Comment (RC2)

2019.

[referee-annotated manuscript omitted]

---

## Author Comment (AC1)

**Author Response to Reviewer Comments:**

**Reviewer 1:**

Thank you for taking the time to review the manuscript. Your overview of the main findings of the paper are consistent with what we had intended to communicate and we are very glad that this came across.

We appreciate your feedback on the figures being hard to read and we will update them accordingly.

In terms of the minor text changes, we added the definition of ABF on page 6 as suggested. We also agree that it is important to clarify that we are talking about boreal spring/summer/fall/winter and we updated the text to reflect this.

**Reviewer 2:**

Thank you as well for taking the time to review the manuscript. We agree with your point that the results also have relevance for PM retrievals and we will add this point to the manuscript and highlight its importance in the abstract and conclusions.

In regards to your question of whether we used MAN data, we did not. Since the MAN data is shipborne and available on a periodic basis, we would not be able to use it to evaluate the long-term AOD and PW relationship at a fixed location in time, as is done in this analysis. However, since AERONET observations are quite limited in marine environments, it is a good point that MAN data is very valuable and would be worth using in another evaluation of the aerosol and water vapor relationship on an event level. A sentence clarifying that this data is not included in the analysis was added to the manuscript.

We agree with the typos that you highlighted in the manuscript and these will be corrected.

In regards to your question on page 5, AERONET AOD is not assimilated in the NAAPS-RA. A sentence clarifying this point was added to the manuscript.

For your comment on page 9, we added a reference to previous manuscripts which include Figures of the same verification regions that are discussed in the manuscript.

For your comment on page 12 and 14, yes, exactly. Many of the sites in which discrepancies were identified were mountainous sites in which we know we don't capture the orography and small scale features in the NAAPS-RA.

For your comment on page 16, yes, as next steps in this work, we are looking at correlations in the vertical using various LIDAR and sounding data. These evaluations will be on a case-by-case basis.

In regards to your comment on page 17 and 22 about PW being a good tracer for AOD but not necessarily aerosol mass, yes, we agree with your point and we will add some text to highlight this point in our conclusions and abstract.

---

## Author Response (AR1)

**Author Response to Reviewer Comments:**

**Reviewer 1:**

- **The reviewer wrote that the figures are very small and details are difficult to read in print version and asked for the labels to be increased in size.**
    - **Response:** we updated the figures to try and make them bigger and easier to read, particularly Figure 7 and 14 which are the central results of the paper. Many of the other figures were increased in size as well. We hope that this makes them more useful.
- **The reviewer asked to introduce the definition of ABF on page 6.**
    - **Response:** We added to line 198, "where $\gamma_i$ is an empirical species-dependent exponent ( anthropogenic/biogenic fine (ABF) ".
- **The reviewer asked to clarify the timeframe for appearance of the negative correlation in spring/summer months (page 11). Is this boreal spring/summer?**
    - **Response: We agree it is important to clarify the time frame. In order to address this, we added the months we are referring to in parentheses on line 383: "**Likewise, stronger European AOD and PW correlations are found in the summer months (JJA), in Eastern Asia in the winter season (DJF), and the Middle East in the fall (SON)." And also added on line 400-401 "During all seasons, negative correlations are found in the Sahel region in both AERONET and the NAAPS-RA with the negative relationships extending further northwards in the boreal spring and summer months."

**Reviewer 2:**

- **Comment on Page 1:** Some types were highlighted in the abstract.
    - **Response:** the highlighted typos were corrected in the abstract accordingly.
- **Comment on Page 1:** The final sentence of the abstract was highlighted and the reviewer wrote "This is also very important for retrievals of PM from space"
    - **Response:** we agree with this statement and updated the final sentence to reflect this: "Given these results, PW can be exploited in coupled aerosol and meteorology data assimilation for AOD and the collocation of aerosol and water vapor should be carefully taken into account when conducting particulate matter (PM) retrievals from space and in evaluating radiative impacts of aerosol, with the season and location in mind. " Also, given the word count limit, we went back through the abstract and updated the wording a little to maintain the word count.
- **Comment on Page 4 (line 143): the reviewer suggested putting a comma instead of "and" in this sentence.**
    - **Response:** We disagree here and kept "and" as aerosol microphysical and radiative properties are grouped together (i.e. aerosol microphysical properties and aerosol radiative properties).
- **Comment on Page 4 (Section 2.1.1 AERONET AOD): The reviewer asked if MAN data was included in the evaluation.**
    - **Response:** We did not include MAN data due to the nature of that data not fitting into our existing analysis, although we agree it would be a useful dataset. We added a sentence at the end of section 2.1.1 to clarify this point: "It should be noted that the AERONET-Maritime Aerosol Network (MAN) data is not included in this analysis as MAN data is shipborne and available on a periodic basis, and thus is not consistent with the long-term evaluation at fixed points that is conducted in this work."

- **Comment on Page 5 (Section 2.1.3 NAAPS Reanalysis):** The reviewer asked in AERONET AOD data is assimilated in the NAAPS-RA.
  - **Response**: It is not. I sentence to clarify this point was added to the manuscript on line 181: "AERONET AOD is not assimilated in the NAAPS-RA".
- **Comment on Page 9 (Section 3.1 Global Patterns of AOD-PW Correlation):** The reviewer suggested adding an additional figure in the appendix to support the region by region discussion.
  - **Response:** Since the regions discussed in this work are consistent with designated verification regions used in previously published work (Figure 4 in Lynch et al. 2016 and Figure 1 Rubin et al. 2016), we chose to add a reference to these papers on line 302-303: "A more in-depth discussion of the data by region, which is consistent with verification regions presented in Lynch et al. (2016) and Rubin et al. (2016), is below:"
- **Comment on Page 12 (Section 3.1 Global Patterns of AOD-PW Correlation):** The reviewer highlighted a sentence "While NAAPS and AERONET are in general agreement in the locations of negative correlations, this discrepancy is likely related to meso or small scale features that are not captured in a global, 1 degree model." And wrote "That is true, particularly for mountainous sites where the difference in model orography and the actual height of the station can be large"
  - **Response**: We took this as the reviewer agreeing with the point we were making. Just for clarification, we added to line 421: "While NAAPS and AERONET are in general agreement in the locations of negative correlations, this discrepancy is likely related to meso or small scale features, including orography, that are not captured in a global, 1 degree model. "
- **Comment on Page 14 (Section 3.3 Slope Evaluation):** The reviewer highlighted the sentence "This terrain/altitude influence is likely a factor in the discrepancies." And wrote "Definitely, it is hard to make meaningful model-obs comparisons at mountain sites due to coarse model resolution and differences in orography".
  - **Response:** We appreciate the reviewer agreeing with our point here. Nothing was changed.
- **Comment on Page 16 (Section 3.5 Vertical Evaluation of the AOD and PW Relationship):** The reviewer highlighted the sentence "This indicates that the model may be transporting too much dust aerosol and water vapor higher into the atmosphere and this transport is well correlated" and wrote "Is this supported by vertically resolved observations such as CALIPSO for aerosol extinction and MW sounders for PW? This is beyond the scope of the paper, but could be interesting for future study.
  - **Response:** We are currently using lidar and sounding data to evaluate these relationships vertically (we do see too much vertical diffusion in the model), but as the reviewer

commented, this is beyond the scope of the paper. We hope to show some of that work in future publications.

- **Comment on Page 17 (Section 3.6 Impact of Hygroscopic Growth): The reviewer highlighted the sentence "**In this highly correlated region, hygroscopic growth is expected to be a significant driver in AOD and PW relationships when dust is not the dominant aerosol type" and wrote "In urban areas this is extremely important as AOD is not a good proxy for dry PM".

    - **Response:** We agree with this point. The hygroscopic growth evaluation helps highlight areas where AOD is not a good proxy for dry PM. We didn't update anything at the highlighted sentence, but we did add a point regarding this in the conclusions, writing on line 812-813 "These findings are also valuable in identifying locations with potential for PM retrieval from space in which hygroscopic growth was found not to be an important factor in the AOD and PW relationship."

- **Comment on Page 22 (Conclusions):** The reviewer highlighted the sentence: "This indicates that PW is a good tracer for AOD, but not necessarily aerosol mass" and wrote "this is an extremely important point.

    - **Response:** We agree and appreciate the reviewer highlighting this as an important point in the paper. We added to this sentence "This indicates that PW is a good tracer for AOD, but not necessarily aerosol mass. This finding has relevance for data assimilation applications as well as PM retrievals." This point is also included in the abstract: "The importance of hygroscopic growth in these relationships indicates that PW is a useful tracer for AOD, or light extinction, but not necessarily as strongly for aerosol mass."